# Continuously Discovering Novel Strategies via Reward-Switching Policy Optimization

**Zihan Zhou**[*†][1♭]**, Wei Fu**[* 2♯]**, Bingliang Zhang**[2]**, Yi Wu**[23♮]

[1] CS Department, University of Toronto, [2] IIIS, Tsinghua University, [3] Shanghai Qi Zhi Institute

[♭] footoredo@gmail.com, [♯] fuwth17@gmail.com, [♮] jxwuyi@gmail.com

## Abstract

We present Reward-Switching Policy Optimization (RSPO), a paradigm to discover diverse strategies in complex RL environments by iteratively finding novel policies that are both locally optimal and sufficiently different from existing ones. To encourage the learning policy to consistently converge towards a previously undiscovered local optimum, RSPO switches between extrinsic and intrinsic rewards via a trajectory-based novelty measurement during the optimization process. When a sampled trajectory is sufficiently distinct, RSPO performs standard policy optimization with extrinsic rewards. For trajectories with high likelihood under existing policies, RSPO utilizes an intrinsic diversity reward to promote exploration. Experiments show that RSPO is able to discover a wide spectrum of strategies in a variety of domains, ranging from single-agent navigation tasks and MuJoCo control to multi-agent stag-hunt games and the StarCraft II Multi-Agent Challenge.

## 1 Introduction

The foundation of deep learning successes is the use of stochastic gradient descent methods to obtain a local minimum for a highly non-convex learning objective. It has been a popular consensus with theoretical justifications that most local optima are very close to the global optimum (Ma, 2020). Consequently, algorithms for most classical deep learning applications only focus on the final performance of the learned local solution rather than *which* local minimum is discovered.

However, this assumption can be problematic in reinforcement learning (RL), where different local optima in the policy space can correspond to substantially different strategies. Therefore, discovering a diverse set of policies can be critical for many RL applications, such as producing natural dialogues in chatbot (Li et al., 2016), improving the chance of finding a targeted molecule (Pereira et al., 2021), generating novel designs (Wang et al., 2019) or training a specialist robot for fast adaptation (Cully et al., 2015). Moreover, in the multi-agent setting, a collection of diverse local optima could further result in interesting emergent behaviors (Liu et al., 2019; Zheng et al., 2020; Baker et al., 2020) and discovery of multiple Nash equilibria (Tang et al., 2021), which further help build strong policies that can adapt to unseen participating agents in a zero-shot manner in competitive (Jaderberg et al., 2019; Vinyals et al., 2019) and cooperative games (Lupu et al., 2021).

In order to obtain diverse strategies in RL, most existing works train a large population of policies in parallel (Pugh et al., 2016; Cully et al., 2015; Parker-Holder et al., 2020b). These methods often adopt a soft learning objective by introducing additional diversity intrinsic rewards or auxiliary losses. However, when the underlying reward landscape in the RL problem is particularly non-uniform, policies obtained by population-based methods often lead to visually identical strategies (Omidshafiei et al., 2020; Tang et al., 2021). Therefore, population-based methods may require a substantially large population size in order to fully explore the policy space, which can be computationally infeasible. Moreover, the use of soft objective also results in non-trivial and subtle hyper-parameter tuning to balance diversity and the actual performance in the environment, which largely prevents these existing methods from discovering *both diverse and high-quality* policies in practice (Parker-Holder et al., 2020b; Lupu et al., 2021; Masood & Doshi-Velez, 2019). Another type of methods directly explores diverse strategies in the reward space by performing multi-objective optimization over

---

[*]Equal Contribution.

[†]Work done as a residence researcher at Shanghai Qi Zhi Institute.

human-designed behavior characterizations (Pugh et al., 2016; Cully et al., 2015) or random search over linear combinations of the predefined objectives (Tang et al., 2021; Zheng et al., 2020; Ma et al., 2020). Although these multi-objective methods are particularly successful, a set of well-defined and informative behavior objectives may not be accessible in most scenarios.

We propose a simple, generic and effective *iterative* learning algorithm, *Reward-Switching Policy Optimization (RSPO)*, for continuously discovering novel strategies under a single reward function without the need of any environment-specific inductive bias. RSPO discovers novel strategies by solving a filtering-based objective, which restricts the RL policy to converge to a solution that is sufficiently different from a set of locally optimal reference policies. After a novel strategy is obtained, it becomes another reference policy for future RL optimization. Therefore, by repeatedly running RSPO, we can quickly derive diverse strategies in just a few iterations. In order to strictly enforce the novelty constraints in policy optimization, we adopt rejection sampling instead of optimizing a soft objective, which is adopted by many existing methods by converting the constraints as Lagrangian penalties or intrinsic rewards. Specifically, RSPO *only* optimizes extrinsic rewards over trajectories that have sufficiently low likelihood w.r.t. the reference policies. Meanwhile, to further utilize those rejected trajectories that are not distinct enough, RSPO ignores the environment rewards on these trajectories and only optimizes diversity rewards to promote effective exploration. Intuitively, this process adaptively switches the training objective between extrinsic rewards and diversity rewards w.r.t. the novelty of each sampled trajectory, so we call it the *Reward Switching* technique.

We empirically validate RSPO on a collection of highly multi-modal RL problems, ranging from multi-target navigation (Mordatch & Abbeel, 2018) and MuJoCo control (Todorov et al., 2012) in the single-agent domain, to stag-hunt games (Tang et al., 2021) and the StarCraft II Multi-Agent Challenge (SMAC) (Rashid et al., 2019) in the multi-agent domain. Experiments demonstrate that RSPO can reliably and efficiently discover surprisingly diverse strategies in all these challenging scenarios and substantially outperform existing baselines. The contributions can be summarized as follows:

1. We propose a novel algorithm, *Reward-Switching Policy Optimization*, for continuously discovering diverse policies. The iterative learning scheme and reward-switching technique both significantly benefit the efficiency of discovering strategically different policies.

2. We propose to use cross-entropy-based diversity metric for policy optimization and two additional diversity-driven intrinsic rewards for promoting diversity-driven exploration.

3. Our algorithm is both general and effective across a variety of single-agent and multi-agent domains. Specifically, our algorithm is the first to learn the optimal policy in the stag-hunt games without any domain knowledge, and successfully discovers 6 visually distinct winning strategies via merely 6 iterations on a hard map in SMAC.

## 2 RELATED WORK

Searching for diverse solutions in a highly multi-modal optimization problem has a long history and various block-box methods have been proposed (Miller & Shaw, 1996; Deb & Saha, 2010; Kroese et al., 2006). In reinforcement learning, one of the most popular paradigms is population-based training with multi-objective optimization. Representative works include the family of qualitative diversity (QD) (Pugh et al., 2016) algorithms, such as MAP-Elites (Cully et al., 2015), which are based on genetic methods and assume a set of human-defined behavior characterizations, and policy-gradient methods (Ma et al., 2020; Tang et al., 2021), which typically assume a distribution of reward function is accessible. There are also some recent works that combine QD algorithms and policy gradient algorithms (Cideron et al., 2020; Nilsson & Cully, 2021). The DvD algorithm (Parker-Holder et al., 2020b) improves QD by optimizing *population diversity (PD)*, a KL-divergence-based diversity metric, without the need of hand-designed behavior characterizations. Similarly, Lupu et al. (2021) proposes to maximize *trajectory diversity*, i.e., the approximated Jensen-Shannon divergence with action-discounting kernel, to train a diversified population.

There are also works aiming to learn policies iteratively. PSRO (Lanctot et al., 2017) focuses on learning Nash equilibrium strategies in zero-sum games by maintaining a strategy oracle and repeatedly adding best responses to it. Various improvements have been made upon PSRO by using different metrics to promote diverse oracle strategies (Liu et al., 2021; Nieves et al., 2021). Hong et al. (2018) utilizes the KL-divergence between the current policy and a past policy version as an exploration bonus, while we are maximizing the diversity w.r.t a fixed set of reference policies,

which is more stable and will not incur a cyclic training process. Diversity-Inducing Policy Gradient (DIPG) (Masood & Doshi-Velez, 2019) utilizes maximum mean discrepancy (MMD) between policies as a soft learning objective to iteratively find novel policies. By contrast, our method utilizes a filtering-based objective via reward switching to strictly enforce all the diversity constraints. Sun et al. (2020) adopts a conceptually similar objective by early terminating episodes that do not incur sufficient novelty. However, Sun et al. (2020) does not leverage any exploration technique for those rejected samples and may easily suffer from low sample efficiency in challenging RL tasks we consider in this paper. There is another concurrent work with an orthogonal focus, which directly optimizes diversity with reward constraints (Zahavy et al., 2021). We remark that enforcing a reward constraint can be problematic in multi-agent scenarios where different Nash Equilibrium can have substantially different pay-offs. In addition, the ridge rider algorithm (Parker-Holder et al., 2020a) proposes to follow the eigenvectors of the Hessian matrix to discover diverse local optima with theoretical guarantees, but Hessian estimates can be extremely inaccurate in complex RL problems.

Another stream of work uses unsupervised RL to discover diverse skills without the use of environment rewards, such as DIYAN (Eysenbach et al., 2019) and DDLUS (Hartikainen et al., 2020). However, ignoring the reward signal can substantially limit the capability of discovering strategic behaviors. SMERL (Kumar et al., 2020) augments DIYAN with extrinsic rewards to induce diverse solutions for robust generalization. These methods primarily focus on learning low-level locomotion while we tackle a much harder problem of discovering strategically and visually different policies.

Finally, our algorithm is also conceptually related to exploration methods (Zheng et al., 2018; Burda et al., 2019; Simmons-Edler et al., 2019), since it can even bypass inescapable local optima in challenging RL environments. Empirical comparisons can be found in Section 4. However, we emphasize that our paper tackles a much more challenging problem than standard RL exploration: we aim to discover as many *distinct local optima* as possible. That is, even if the global optimal solution is discovered, we still want to continuously seek for sufficiently distinct local-optimum strategies. We remark that such an objective is particularly important for multi-agent games where finding all the Nash equilibria can be necessary for analyzing rational multi-agent behaviors (Tang et al., 2021).

## 3 METHOD

### 3.1 PRELIMINARY

We consider environments that can be modeled as a Markov decision process (MDP) (Puterman, 1994) $M = (\mathcal{S}, \mathcal{A}, R, P, \gamma)$ where $\mathcal{S}$ and $\mathcal{A}$ are the state and action space respectively, $R(s, a)$ is the reward function, $P(s'|s, a)$ is the transition dynamics and $\gamma$ is the discount factor. We consider a stochastic policy $\pi_\theta$ paramterized by $\theta$. Reinforcement learning optimizes the policy w.r.t. the expected return $J(\pi) = \mathbb{E}_{\tau \sim \pi}[\sum_t \gamma^t r_t]$ over the sampled trajectories from $\pi$, where a trajectory $\tau$ denotes a sequence of state-action-reward triplets, i.e., $\tau = \{(s_t, a_t, r_t)\}$. Note that this formulation can be naturally applied to multi-agent scenarios with homogeneous agents with shared state and action space, where learning a shared policy for all the agents will be sufficient.

Rather than learning a single solution for $J(\theta)$, we aim to discover a diverse set of $M$ policies, i.e, $\{\pi_{\theta^k} | 1 \le k \le M\}$, such that all of these polices are locally optimized under $J(\theta)$ and mutually distinct w.r.t. some distance measure $D(\pi_{\theta^i}, \pi_{\theta^j})$, i.e.,

$$\max_{\theta^k} J(\theta^k) \ \forall 1 \le k \le M, \quad \text{subject to } D(\pi_{\theta^i}, \pi_{\theta^j}) \ge \delta, \quad \forall 1 \le i < j \le M. \tag{1}$$

Here $D(\cdot, \cdot)$ measures how different two policies are and $\delta$ is the novelty threshold. For conciseness, in the following content, we omit $\theta$ and use $\pi_k$ to denote the policy with parameter $\theta^k$.

### 3.2 ITERATIVE CONSTRAINED POLICY OPTIMIZATION

Directly solving Eq. (1) suggests a population-based training paradigm, which requires a non-trivial optimization technique for the pairwise constraints and typically needs a large population size $M$. Herein, we adopt an iterative process to discover novel policies: in the $k$-th iteration, we optimize a single policy $\pi_k$ with the constraint that $\pi_k$ is sufficiently distinct from previously discovered policies $\pi_1, \ldots, \pi_{k-1}$. Here, the term "iteration" is used to denote the process of learning a new policy. Formally, we solve the following iterative constrained optimization problem for iteration $1 \le k \le M$:

$$\theta_k = \arg \max_\theta J(\theta), \quad \text{subject to } D(\pi_\theta, \pi_j) \ge \delta, \quad \forall 1 \le j < k. \tag{2}$$

Eq. (2) reduces the population-based objective to a standard constrained optimization problem for a single policy, which is much easier to solve. Such an iterative procedure does not require a large population size $M$ as is typically necessary in population-based methods. And, in practice, only a few iterations could result in a sufficiently diverse collection of policies. We remark that, in theory, directly solving the constraint problem in Eq. (2) may lead to a solution that is not a local optimum w.r.t. the unconstrained objective $J(\theta)$. It is because a solution in Eq. (2) can be located on the boundary of the constraint space (i.e., $D(\pi_\theta, \pi_j) = \delta$), which is undesirable according to our original goal. However, this issue can be often alleviated by properly setting the novelty threshold $\delta$.

The natural choice for measuring the policy difference is KL divergence, as done in the trust-region constraint (Schulman et al., 2015; 2017). However, in our setting where the difference between policies should be maximized, using KL as the diversity measure would inherently encourage learning a policy with small entropy, which is typically undesirable in RL problems (see App. F for a detailed derivation). Therefore, we adopt the accumulative cross-entropy as our diversity measure, i.e.,

$$D(\pi_i, \pi_j) := \mathcal{H}(\pi_i, \pi_j) = \mathbb{E}_{\tau \sim \pi_i} \left[ - \sum_t \log \pi_j(a_t \mid s_t) \right] \tag{3}$$

### 3.3 TRAJECTORY FILTERING FOR ENFORCING DIVERSITY CONSTRAINTS

A popular approach to solve the constrained optimization problem in Eq. (2) is to use Lagrangian multipliers to convert the constraints to penalties in the learning objective. Formally, let $\beta_1, \ldots, \beta_{k-1}$ be a set of hyperparameters, the soft objective for Eq. (2) is defined by

$$J_{\text{soft}}(\theta) := J(\pi_\theta) + \sum_{j=1}^{k-1} \beta_j D(\pi_\theta, \pi_j). \tag{4}$$

Such a soft objective substantially simplifies optimization and is widely adopted in RL applications. However, in our setting, since cross-entropy is a particularly dense function, including the diversity bonus as part of the objective may largely change the reward landscape of the original RL problem, which could make the final solution diverge from a locally optimal solution w.r.t $J(\theta)$. Therefore, it is often necessary to anneal the Lagrangian multipliers $\beta_j$, which is particularly challenging in our setting with a large number of reference policies. Moreover, since $D(\pi_\theta, \pi_j)$ is estimated over the trajectory samples, it introduces substantially high variance to the learning objective, which becomes even more severe as more policies are discovered.

Consequently, we propose a *Trajectory Filtering* objective to alleviate the issues of the soft objective. Let's use $\text{NLL}(\tau; \pi)$ to denote the negative log-likelihood of a trajectory $\tau$ w.r.t. a policy $\pi$, i.e., $\text{NLL}(\tau; \pi) = - \sum_{(s_t, a_t) \sim \tau} \log \pi(a_t | s_t)$. We apply rejection sampling over the sampled *trajectories* $\tau \sim \pi_\theta$ such that we train on those trajectories satisfying *all* the constraints, i.e., $\text{NLL}(\tau; \pi_j) \geq \delta$ for each reference policy $\pi_j$. Formally, for each sampled trajectory $\tau$, we define a filtering function $\phi(\tau)$, which indicates whether we want to reject the sample $\tau$, and use $\mathbb{I}[\cdot]$ to denote the indicator function, and then the trajectory filtering objective $J_{\text{filter}}(\theta)$ can be expressed as

$$J_{\text{filter}}(\theta) = \mathbb{E}_{\tau \sim \pi_\theta} \left[ \phi(\tau) \sum_t \gamma^t r_t \right], \quad \text{where} \ \ \phi(\tau) := \prod_{j=1}^{k-1} \mathbb{I}[\text{NLL}(\tau; \pi_j) \geq \delta]. \tag{5}$$

We call the objective in Eq. (5) a *filtering* objective. We show in App. G that solving Eq. (5) is equivalent to solving Eq. (2) with an even stronger diversity constraint. In addition, we also remark that trajectory filtering shares a conceptually similar motivation with the clipping term in Proximal Policy Optimization (Schulman et al., 2017).

### 3.4 INTRINSIC REWARDS FOR DIVERSITY EXPLORATION

The main issue in Eq. (5) is that trajectory filtering may reject a significant number of trajectories, especially in the early stage of policy learning since the policy is typically initialized to the a random policy. Hence, it is often the case that most of the data in a batch are abandoned, which leads to a severe wasting of samples and may even break learning due to the lack of feasible trajectories.

*Can we make use of those rejected trajectories?* We propose to additionally apply a novelty-driven objective on those rejected samples. Formally, we use $\phi_j(\tau)$ to denote whether $\tau$ violates the constraint of $\pi_j$, i.e., $\phi_j(\tau) = \mathbb{I}[\text{NLL}(\tau, \pi_j) \geq \delta]$. Then we have the following *switching* objective:

$$J_{\text{switch}} = \mathbb{E}_{\tau \sim \pi_\theta} \left[ \phi(\tau) \sum_t \gamma^t r_t + \lambda \sum_j (1 - \phi_j(\tau)) \, \text{NLL}(\tau, \pi_j) \right] \quad (6)$$

The above objective simultaneously maximizes the extrinsic return on accepted trajectories and the cross-entropy on rejected trajectories. It can be proved that solving Eq. (6) is also equivalent to solving Eq. (2) with a stronger diversity constraint (see App. G).

Furthermore, Eq. (6) can be also interpreted as introducing additional cross-entropy intrinsic rewards on rejected trajectories (i.e., $\phi_j(\tau) = 0$). More specifically, given $\text{NLL}(\tau; \pi) = -\sum_{(s_t, a_t) \in \tau} \log \pi(a_t | s_t)$, an intrinsic reward $r^{\text{int}}(s_t, a_t; \pi_j) = -\log \pi_j(a_t \mid s_t)$ is applied to each state-action pair $(s_t, a_t)$ from every rejected trajectory $\tau$. Conceptually, this suggests an even more general paradigm for diversity exploration: we can optimize extrinsic rewards on accepted trajectories while utilizing novelty-driven intrinsic rewards on rejected trajectories for more effective exploration, i.e., by encouraging the learning policy $\pi_\theta$ to be distinct from a reference policy $\pi_j$.

Hence, we propose two different types of intrinsic rewards to promote diversity exploration: one is *likelihood-based*, which directly follows Eq. (6) and focuses more on behavior novelty, and the other is *reward-prediction-based*, which focuses more on achieving novel states and reward signals.

**Behavior-driven exploration.**   The behavior-driven intrinsic reward $r_{\mathbf{B}}^{\text{int}}$ is defined by

$$r_{\mathbf{B}}^{\text{int}}(a, s; \pi_j) = -\log \pi_j(a \mid s). \quad (7)$$

$r_{\mathbf{B}}^{\text{int}}$ encourages the learning policy to output different actions from those reference policies and therefore to be more likely to be accepted. Note that $r_{\mathbf{B}}^{\text{int}}$ can be directly interpreted as the Lagrangian penalty utilized in the soft objective $J_{\text{soft}}(\theta)$.

**Reward-driven exploration.**   A possible limitation of behavior-driven exploration is that it may overly focus on visually indistinguishable action changes rather than high-level strategies. Note that in RL problems with diverse reward signals, it is usually preferred to discover policies that can achieve different types of rewards (Simmons-Edler et al., 2020; Wang* et al., 2020). Inspired by the curiosity-driven exploration method (Pathak et al., 2017), we adopt a model-based approach for predicting novel reward signals. In particular, after obtaining each reference policy $\pi_j$, we learn a reward prediction function $f(s, a; \psi_j)$ trained by minimizing the expected MSE loss $\mathcal{L}(\psi_j) = \mathbb{E}_{\tau \sim \pi_j, t} \left[ |f(s_t, a_t; \psi_j) - r_t|^2 \right]$ over the trajectories generated by $\pi_j$. The reward prediction function $f(s, a; \psi_j)$ is expected to predict the extrinsic environment reward more accurately on state-action pairs that are more frequently visited by $\pi_j$ and less accurately on rarely visited pairs. To encourage policy exploration, we adopt the reward prediction error as our reward-driven intrinsic reward $r_{\mathbf{R}}^{\text{int}}(a, s; \pi_j)$. Formally, given the transition triplet $(s_t, a_t, r_t)$, $r_{\mathbf{R}}^{\text{int}}(a, s; \pi_j)$ is defined by

$$r_{\mathbf{R}}^{\text{int}}(s_t, a_t; \pi_j) = |f(s_t, a_t; \psi_j) - r_t|^2. \quad (8)$$

We remark that reward-driven exploration can be also interpreted as approximately maximizing the $f$-divergence of joint state occupancy measure between policies (Liu et al., 2021). By combining these two intrinsic rewards together, we approximately maximize the divergence of *both actions and states* between policies to effectively promote diversity. By default, we use behavior-driven intrinsic reward for computational simplicity and optionally augment it with reward-driven intrinsic reward in more challenging scenarios (see examples in Section 4.2).

## 3.5   REWARD-SWITCHING POLICY OPTIMIZATION

We define the RSPO function $r_t^{\text{RSPO}}$ by

$$r_t^{\text{RSPO}} = \phi(\tau) r_t + \lambda \sum_j (1 - \phi_j(\tau)) r^{\text{int}}(a_t, s_t; \pi_j), \quad (9)$$

where $\lambda$ is a scaling hyper-parameter. Note that extrinsic rewards and intrinsic rewards are mutually exclusive, i.e., a trajectory $\tau$ may be either included in $J_{\text{filtering}}$ or be rejected to produce exploration bonuses. Conceptually, our method is adaptively "switching" between extrinsic and intrinsic rewards during policy gradients, which is so-called *Reward-Switching Policy Optimization (RSPO)*. We also remark that the intrinsic reward will constantly push the learning policy towards the feasible policy space and the optimization objective will eventually converge to $J(\theta)$ when no trajectory is rejected.

In addition to the aforementioned RSPO algorithm, we also introduce two implementation enhancements for better empirical performances, especially in some performance-sensitive scenarios.

**Automatic threshold selection.**   We provide an empirical way of adjusting $\delta$. In some environments, $\delta$ is sensitive to each reference policy. Instead of tuning $\delta$ for each reference policy, we choose its corresponding threshold by $\delta_j = \alpha \cdot D(\pi^{\mathrm{rnd}}, \pi_j)$, where $\pi^{\mathrm{rnd}}$ is a fully random policy and $\alpha$ is a task-specific hyperparameter. We remark that $\alpha$ is a constant parameter across training iterations and is much easier to choose than manually tuning $\delta$, which requires subtle variation throughout multiple training iterations. We use automatic threshold selection by default. Detailed values of $\alpha$ and the methodology of tuning $\alpha$ can be found in App. D.1 and App. B.3 respectively.

**Smoothed-switching for intrinsic rewards.**   Intrinsic rewards have multiple switching indicators, i.e., $\phi_1, \ldots, \phi_{k-1}$. Moreover, for different trajectories, different subsets of indicators will be turned on and off, which may result in a varying scale of intrinsic rewards and hurt training stability. Therefore, in some constraint-sensitive cases, we propose a smoothed switching mechanism which could further improve practical performance. Specifically, we maintain a running average $\tilde{\phi}_j$ over all the sampled trajectories for each indicator $\phi_j(\tau)$, and use these smoothed indicators to compute intrinsic rewards defined in Eq. (9). Smoothed-switching empirically improves training stability when a large number of reference policies exist, such as in stag-hunt games (see Section 4.2).

## 4    EXPERIMENTS

To illustrate that our method can be applied to *general* RL applications, we experiment on 4 domains that feature multi-modality of solutions, including a single-agent navigation problem in the particle-world (Mordatch & Abbeel, 2018), 2-agent Markov stag-hunt games (Tang et al., 2021), continuous control in MuJoCo (Todorov et al., 2012), and the StarCraft II Multi-Agent Challenge (SMAC) (Vinyals et al., 2017; Rashid et al., 2019). In particle world and stag-hunt games, all the local optima can be precisely calculated, so we can quantitatively evaluate the effectiveness of different algorithms by measuring how many distinct strategy modes are discovered. In MuJoCo control and SMAC, we qualitatively demonstrate that our method can discover a large collection of visually distinguishable strategies. Notably, we primarily present results from purely RL-based methods which do not require prior knowledge over possible local optima for a fair comparison. We also remark that when a precise feature descriptor of local optima is feasible, it is also possible to apply evolutionary methods (Nilsson & Cully, 2021) to a subset of the scenarios we considered. For readers of further interest, a thorough study with discussions can be found in App. B.4.

Our implementation is based on PPO (Schulman et al., 2017) on a desktop machine with one CPU and one NVIDIA RTX3090 GPU. All the algorithms are run for the same number of total environment steps and the same number of iterations (or population size). More details can be found in appendix.

### 4.1    SINGLE-AGENT PARTICLE-WORLD ENVIRONMENT

We consider a sparse-reward navigation scenario called *4-Goals* (Fig. 1). The agent starts from the center and will receive a reward when reaching a landmark. We set up 3 difficulty levels. In the easy mode, the landmark locations are fixed. In the medium mode, the landmarks are randomly placed. In the hard mode, landmarks are not only placed randomly but also have different sizes and rewards. Specifically, the sizes and rewards of each landmark are $2\times, 1\times, 0.5\times, 0.25\times$ and $1\times, 1.1\times, 1.2\times,$

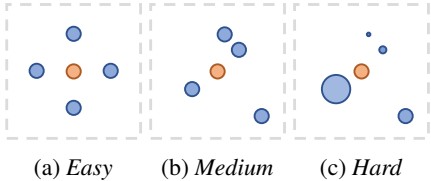

(a) *Easy*   (b) *Medium*   (c) *Hard*

Figure 1: The agent (orange) and landmarks (blue) in *4-Goals*.

$1.3\times$ of the normal one respectively. We remark that in the hard mode, the landmark size decreases at an exponential rate while the reward gain is only marginal, making it exponentially harder to discover policies towards those smaller landmarks. We compare RSPO with several baselines, including PPO with restarts (PG), Diversity-Inducing Policy Gradient (DIPG) (Masood & Doshi-Velez, 2019), population-based training with cross-entropy objective (PBT-CE), DvD (Parker-Holder et al., 2020b), SMERL (Kumar et al., 2020), and Random Network Distillation (RND) (Burda et al., 2019). RND is designed to explore the policy with the highest reward, so we only evaluate RND in the hard mode.

The number of distinct local optima discovered by different methods is presented in Fig. 2a. RSPO consistently discovers all the 4 modes within 4 iterations even without the use of any intrinsic rewards over rejected trajectories (i.e., $r_t^{\mathrm{int}} = 0$). DIPG finds 4 strategies in 4 out of the 5 runs in the easy mode but performs no better than PG in the two harder modes. Fig. 2b shows the highest expected return achieved over the policy population in the hard mode. RSPO is the only algorithm that

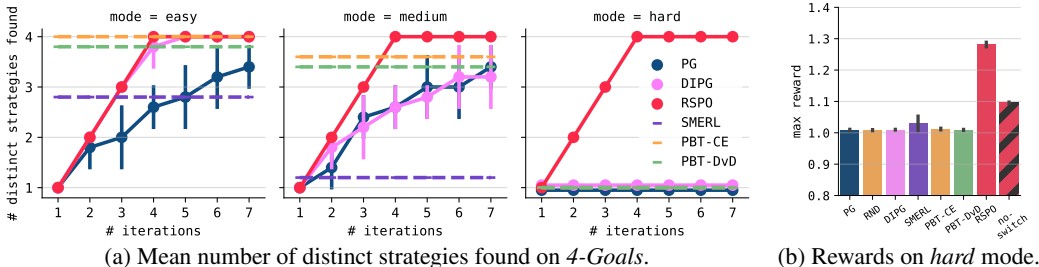

(a) Mean number of distinct strategies found on *4-Goals*.     (b) Rewards on *hard* mode.

Figure 2: Experiment results on *4-Goals* for $M = 7$ iterations averaged over 5 random seeds. Error bars are $95\%$ confidence intervals.

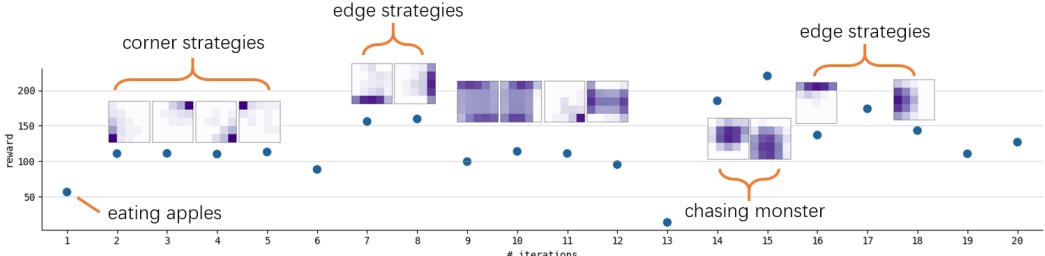

Figure 4: Different strategies found in a run of 20 iterations diversity setting RSPO in *Monster-Hunt*. We plot the heatmap of the two agents' meeting point to indicate the type of the found strategies.

successfully learns the optimal policy towards the smallest ball. We also report the performance of RSPO optimizing the soft objective in Eq. (4) with the default behavior-driven intrinsic reward $r_{\mathbf{B}}^{\text{int}}$ (*no-switch*), which was able to discover the policy towards the second largest landmark. Comparing *no-switch* with $r_t^{int} = 0$, we could conclude that the filtering-based objective can be critical for RSPO to discover sufficiently different modes. We also remark that the *no-switch* variant only differs from DIPG by the used diversity metric. This suggests that in environments with high state variance (e.g. landmarks with random sizes and positions), state-based diversity metric may be less effective.

## 4.2 2-AGENT MARKOV STAG-HUNT GAMES

We further show the effectiveness of RSPO on two grid-world stag-hunt games developed in Tang et al. (2021), *Monster-Hunt* and *Escalation*, both of which have very distinct Nash Equilibria (NEs) for self-play RL methods to converge to. Moreover, the optimal NE with the highest rewards for both agents in these games are *risky cooperation*, i.e., a big penalty will be given to an agent if the *other* agent stops cooperation. This makes most self-play RL algorithms converge to the safe non-cooperative NE strategies with lower rewards. It has been shown that *none* of the

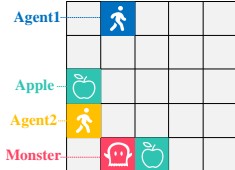

Figure 3: *Monster-Hunt*

state-of-the-art exploration methods can discover the global optimal solution without knowing the underlying reward structure (Tang et al., 2021). We remark that since there are enormous NEs in these environments as shown in Fig. 4 and 7b, population-based methods (PBT) require a significantly large population size for meaningful performances, which is computationally too expensive to run. Therefore, we do not include the results of PBT baselines. We also apply the *smoothed-switching* heuristic for RSPO in this domain. Environment details can be found in App. C.2.

**The *Monster-Hunt* game.** The *Monster-Hunt* game (Fig. 3) contains a monster and two apples. When a single agent meets the monster, it gets a penalty of $-2$. When both agents meet the monster at the same time, they "catch" the monster and both get a bonus of $5$. When a player meets an apple, it gets a bonus of $2$. The optimal strategy, i.e., both agents move towards the monster, is a risky cooperative NE since an agent will receive a penalty if the other agent deceives. The non-cooperative NE for eating apples is a safe NE and easy to discover but has lower rewards.

We adopt both behavior-driven and reward-driven intrinsic rewards in RSPO to tackle *Monster-Hunt*. Fig. 4 illustrates all the discovered strategies by RSPO over 20 iterations, which covers a wide range of human-interpretable strategies, including the non-cooperative apple-eating strategy as well as the

Table 1: Types of strategies discovered by each methods in *Monster-Hunt* over 20 iterations.

| | | Apple | Corner | Edge | Chase |
|---|---|---|---|---|---|
| Ablation | RSPO | ✓ | ✓ | ✓ | ✓ |
| | - No switch | ✓ | ✓ | | |
| | - No $r^{\text{int}}$ | ✓ | | | |
| | - $r_{\mathbf{B}}^{\text{int}}$ only | ✓ | ✓ | ✓ | |
| Baseline | RPG | ✓ | ✓ | | ✓ |
| | MAVEN | ✓ | ✓ | | |
| | PG/DIPG/RND | ✓ | | | |

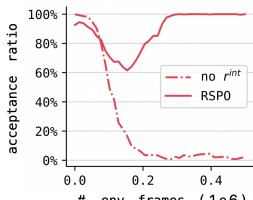

Figure 5: Sample acceptance ratio when learning a policy distinct from *Apple* NE. The intrinsic reward is critical.

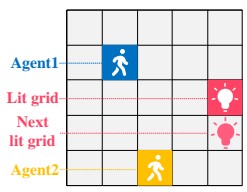

Figure 6: *Escalation*: two agents need to keep stepping on the light simultaneously.

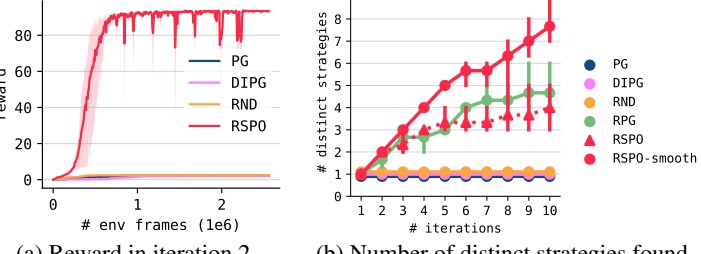

(a) Reward in iteration 2.     (b) Number of distinct strategies found.

Figure 7: Results on *Escalation* averaged over 3 random seeds. Shaded area and error bars are $95\%$ confident intervals.

optimal strategy, where both two agents stay together and chase the monster actively. By visualizing the heatmap of where both agents meet, we observe a surprisingly diverse sub-optimal cooperation NEs, where both agents move to a corner or an edge simultaneously, keep staying there, and wait for the monster coming. We remark that due to the existence of such a great number of passive waiting strategies, which all have similar accumulative environment rewards and states, it becomes critical to include the reward-driven intrinsic reward to quickly bypass them and discover the optimal solution.

We perform ablation studies on RSPO by turning off reward switching (*No Switch*) or intrinsic reward (*No $r^{int}$*) or only using the behavior-driven intrinsic reward ($r_{\mathbf{B}}^{int}$ *only*), and evaluate the performances of many baseline methods, including vanilla PG with restarts (PG), DIPG, RND, a popular multi-agent exploration method MAVEN (Mahajan et al., 2019) and reward-randomized policy gradient (RPG) (Tang et al., 2021). We summarize the categories of discovered strategies by all these baselines in Table 1. *Apple* denotes the non-cooperative apple-eating NE; *Chase* denotes the optimal NE where both agents actively chase the monster; *Corner* and *Edge* denote the sub-optimal cooperative NE where both agents passively wait for the monster at a corner or an edge respectively. Regarding the baselines, PG, DIPG, and RND never discover any strategy beyond the non-cooperative *Apple* NE. For RPG, even using the domain knowledge to change the reward structure of the game, it never discovers the *Edge* NE. Regarding the RSPO variants, both reward switching and intrinsic rewards are necessary. Fig. 5 shows that when the intrinsic reward is turned off, the proportion of accepted trajectories per batch stays low throughout training. This implies that the learning policy failed to escape the infeasible subspace. Besides, as shown in Table 1, using behavior-driven exploration alone fails to discover the optimal NE, which suggests the necessity of reward-driven exploration to maximize the divergence of both states and actions in problems with massive equivalent local optima.

**The *Escalation* game.** *Escalation* (Fig. 6) requires the two players to interact with a static *light*. When *both* players step on the light simultaneously, they both receive a bonus of $1$. Then the light moves to a random adjacent grid. The game continues only if both players choose to follow the light. If only one player steps on the light, it receives a penalty of $-0.9L$, where $L$ is the number of previous cooperation steps. For each integer $L$, there is a corresponding NE where both players follow the light for $L$ steps then simultaneously stop cooperation. We run RSPO with both diversity-driven intrinsic rewards and compare it with PG, DIPG, RND and RPG. Except for RPG, none of the baseline methods discover any cooperative NEs while RSPO directly learns the optimal cooperative NE (i.e., always cooperate) in the second iteration as shown in Fig. 7a. We also measure the total number of discovered NEs by different methods over 10 iterations in Fig. 7b. Due to the existence of many spiky local optima, the *smoothed-switching* technique can be crucial here to stabilize the

Table 2: *Population Diversity* scores in *MuJoCo*.

| | H.-Cheetah | Hopper | Walker2d | Humanoid |
|---|---|---|---|---|
| PG | 0.033 (0.013) | 0.418 (0.125) | 0.188 (0.079) | 0.965 (0.006) |
| DIPG | 0.051 (0.009) | 0.468 (0.054) | 0.179 (0.056) | 0.996 (0.000) |
| PBT-CE | 0.160 (0.078) | 0.620 (0.294) | 0.512 (0.032) | **0.999 (0.000)** |
| DvD | 0.275 (0.164) | 0.656 (0.523) | 0.542 (0.103) | **1.000 (0.000)** |
| SMERL | 0.003 (0.002) | 0.674 (0.389) | 0.669 (0.152) | N/A |
| RSPO | **0.359 (0.058)** | **0.989 (0.009)** | **0.955 (0.039)** | **0.999 (0.000)** |

Table 3: Number of visually distinct policies over 4 iterations in SMAC.

| | 2c64zg | 2m1z |
|---|---|---|
| PG | 2 | 2 |
| DIPG | 2 | 2 |
| PBT-CE | 2 | 3 |
| TrajDiv | 3 | 1 |
| RSPO | **4** | **4** |

training process. We remark that even without the *smoothed-switching* technique, RSPO achieves comparable performance with RPG — note that RPG requires a known reward function while RSPO does not assume any environment-specific domain knowledge.

### 4.3 CONTINUOUS CONTROL IN *MuJoCo*

We evaluate RSPO in the continuous control domain, including *Half-Cheetah*, *Hopper*, *Walker2d* and *Humanoid*, and compare it with baseline methods including PG, DIPG, DvD (Parker-Holder et al., 2020b), SMERL (Kumar et al., 2020) and population-based training with our cross-entropy objective (PBT-CE). All the methods are run over 5 iterations (a population size of 5, or have a latent dimension of 5) across 3 seeds. We adopt *Population Diversity*, a determinant-based diversity criterion proposed in Parker-Holder et al. (2020b), to evaluate the diversity of derived policies by different methods. Results are summarized in Table 2, where RSPO achieves comparable performance in *Hopper* and *Humanoid* and substantially outperforms all the baselines in *Half-Cheetah* and *Walker2d*. We remark that even with the same intrinsic reward, population-based training (PBT-CE) cannot discover sufficiently novel policies compared with iterative learning (RSPO). In *Humanoid*, SMERL achieves substantially lower return than other baseline methods and we don't report the population diversity score (more details can be found in App. B.6). We also visualize some interesting emergent behaviors RSPO discovered for *Half-Cheetah* and *Hopper* in App. B.1, where different strategy modes discovered by RSPO are visually distinguishable while baselines methods often converge to very similar behaviors despite of the non-zero diversity score.

### 4.4 STACRAFT MULTI-AGENT CHALLENGE

We further apply RSPO to the StarCraft II Multi-Agent Challenge (SMAC) (Rashid et al., 2019), which is substantially more difficult due to partial observability, long horizon, and complex state/action space. We conduct experiments on 2 maps, an easy map *2m_vs_1z* and a hard map *2c_vs_64zg*, both of which have heterogeneous unit types leading to a multi-modal solution space. Baseline methods include PG, DIPG, PBT-CE and trajectory diversity (TrajDiv) (Lupu et al., 2021). SMERL algorithm and DvD algorithm are not included because they were originally designed for continuous control domain and not suitable for SMAC (see App. B.6 and App. B.2.2). Instead, we include the TrajDiv algorithm (Lupu et al., 2021) as an additional baseline, which was designed for cooperative multi-agent games. We compare the number of visually distinct policies by training a population of 4 (or for 4 iterations), as shown in Table 3. While PBT-based algorithms tend to discover policies with slight distinctions, RSPO can effectively discover different *winning* strategies demonstrating intelligent behaviors in just a few iterations consistently across repetitions. We remark that there may not exist an appropriate quantitative diversity metric for such a sophisticated MARL game in the existing literature (see App. B.2.2). Visualizations and discussions can be found in App. B.1.

## 5 CONCLUSION

We propose *Reward-Switching Policy Optimization (RSPO)*, a simple, generic, and effective iterative learning algorithm that can continuously discover novel strategies. RSPO tackles a novelty-constrained optimization problem via adaptive switching between extrinsic and intrinsic rewards used for policy learning. Empirically, RSPO can successfully tackle a wide range of challenging RL domains under both single-agent and multi-agent settings. We leave further theoretical justifications and sample efficiency improvements as future work.

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

## A GIF DEMONSTRATIONS ON MUJOCO AND SMAC

See `https://sites.google.com/view/rspo-iclr-2022`.

## B ADDITIONAL RESULTS

### B.1 VISUALIZATION OF DISCOVERED STRATEGIES

#### B.1.1 MUJOCO

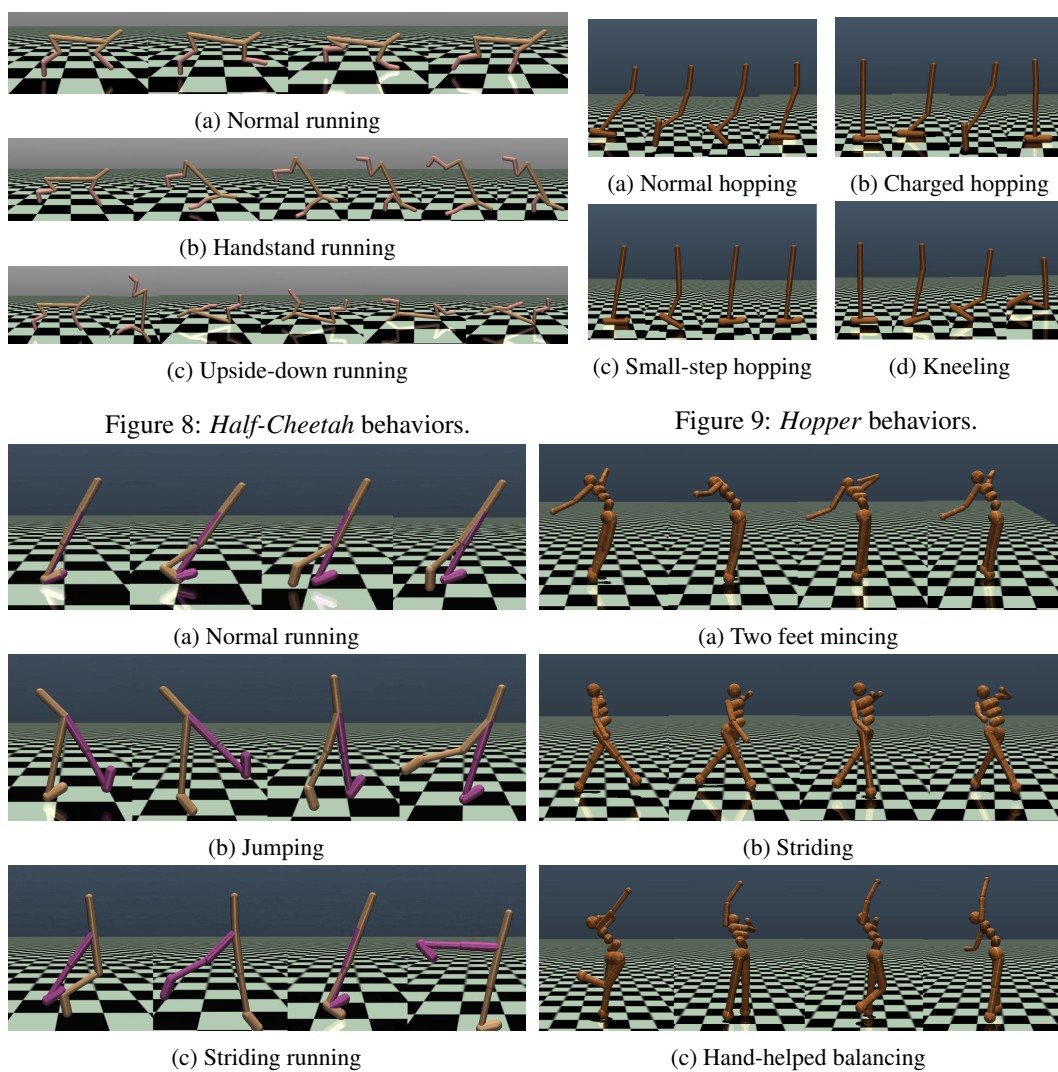

(a) Normal running

(b) Handstand running

(c) Upside-down running

Figure 8: *Half-Cheetah* behaviors.

(a) Normal hopping   (b) Charged hopping

(c) Small-step hopping   (d) Kneeling

Figure 9: *Hopper* behaviors.

(a) Normal running

(b) Jumping

(c) Striding running

Figure 10: *Walker* behaviors.

(a) Two feet mincing

(b) Striding

(c) Hand-helped balancing

Figure 11: *Homanoid* behaviors.

#### B.1.2 SMAC

We present screenshots of emergent strategies in the SMAC environment in Fig. 14 and Fig. 15 for *2c_vs_64zg* and *2m_vs_1z* respectively. We expect different emergent strategies to be both *visually distinguishable* and *human-interpretable*. The strategies induced by baseline methods and RSPO are summarized in Table 4 and Table 5.

On the hard map *2c_vs_64zg*, agents need to control the 2 colossi to fight against 64 zergs. The colossi have a wider attack range and can step over the cliff. Fig. 14a shows an aggressive strategy

where the two colossi keep staying together on the left side of the cliff to fire at all the coming enemies. Fig. 14b shows a particularly intelligent strategy where the colossi make use of the terrain to play hit-and-run. When the game starts, the colossi stand on the cliff to snipe distant enemies to make them keep wandering around under the cliff. Hence, as the game proceeds, we can observe that the enemies are clearly partitioned while the colossi always maintain a smart fire position on the cliff. Fig. 14c shows a mirror strategy similar to Fig. 14a to aggressively clean up incoming enemies from the right side. Fig. 14d shows a conservative strategy that the colossi stand still in the corner to keep minimal contact with the enemies, and thus minimize damages received from enemies. Fig. 14e shows another smart strategy: one colossi blocks all incoming enemies on the mountain pass as a fire attractor, while the other one hides behind the fire attractor and snipes enemies distantly. We can see from the last two frames that the distant sniper does not lose any health points in late stages. In Fig. 14f, one colossi (#1) actively take advantage of the terrain to walk along the cliff, such that enemies on the plateau must run around to attack it. In the mean time, the colossi helps its teammate by sniping distant enemies. Finally, they separately clean up all the remaining enemies for the win.

On the easy map *2m_vs_1z*, agents need to control 2 marines to defeat a Zealot. The marines can shoot the Zealot in a distant position but the Zealot can only perform close-range attacks. In Fig. 15a, the marines swing horizontally to keep an appropriate fire distance from the Zealot. In Fig. 15b, the marines perform a parallel hit-and-run from top to bottom as the Zealot approaches. In Fig. 15c, the right-side marine stands still, and the left-side marine swings vertically to distract the Zealot. In Fig. 15d, the two marines alternatively perform hit-and-run from bottom to top to distract the Zealot.

Table 4: Strategies induced by baseline methods and RSPO in SMAC map *2c_vs_64zg*.

| Algorithm | Strategies |
|---|---|
| PG | left wave cleanup, right wave cleanup |
| DIPG | cliff walk, corner |
| PBT-CE | left wave cleanup, right wave cleanup |
| TrajDiv | fire attractor and distant sniper, cliff walk, right wave cleanup |
| RSPO | left wave cleanup, cliff sniping and smart blocking, right wave cleanup corner, fire attractor and distant sniper, cliff walk |

Table 5: Strategies induced by baseline methods and RSPO in SMAC map *2m_vs_1z*.

| Algorithm | Strategies |
|---|---|
| PG | parallel hit-and-run, alterative distraction |
| DIPG | parallel hit-and-run, one-sided swinging |
| PBT-CE | one-sided swinging, parallel-hit-and-run, swinging |
| TrajDiv | parallel hit-and-run |
| RSPO | one-sided swinging, parallel-hit-and-run, swinging, alterative distraction |

## B.2 QUANTITATIVE EVALUATION

### B.2.1 MUJOCO

The final performance of MuJoCo environments is presented in Table 6. As mentioned in Appendix C.3, in our implementation we fix the episode length to $512$ so that the diverse intrinsic rewards can be easily computed, which may harm sample efficiency and the evaluation score. Moreover, we use a hidden size of $64$ which is usually smaller than previous works (which is $256$ in the SAC paper (Haarnoja et al., 2018)). Hence, these results may not be directly compared with other numbers in the existing literature. However, the policy in Iter #1 is obtained by vanilla PPO and can therefore be used to assess the relative performances of other runs. Note that even if the presented scores are not the state-of-the-art, visualization results show that our algorithm successfully learns the basic locomotion and diverse gaits in these environments. The results indeed demonstrate diverse local optima that are properly discovered by RSPO, including many interesting emergent behaviors

Table 6: Final evaluation performance of RSPO averaged over 32 episodes in MuJoCo continuous control domain. Averaged over 3 random seeds with standard deviation in the brackets.

| Environment | Iter #1 | Iter #2 | Iter #3 | Iter #4 | Iter #5 |
|---|---|---|---|---|---|
| *Half-Cheetah* | 2343 (882) | 1790 (950) | 1710 (1394) | 859 (498) | 1194 (640) |
| *Hopper* | 1741 (56) | 1690 (29) | 1428 (322) | 1349 (275) | 841 (296) |
| *Walker2d* | 1668 (284) | 1627 (228) | 1258 (538) | 1275 (342) | 1040 (318) |
| *Homanoid* | 3146 (26) | 3148 (140) | 3118 (84) | 3072 (76) | 2707 (860) |

Table 7: Final evaluation winning rate of RSPO averaged over 32 episodes in SMAC.

| Map | Iter #1 | Iter #2 | Iter #3 | Iter #4 | Iter #5 | Iter#6 |
|---|---|---|---|---|---|---|
| *2c_vs_64zg* | 100% | 84.4% | 96.9% | 100% | 87.5% | 100% |
| *2m_vs_1z* | 100% | 100% | 100% | 100% | N/A | N/A |

that may have never been reported in the literature (check the website in Appendix A for details), which accords with our initial motivation.

### B.2.2  SMAC

Final evaluation winning rate of SMAC is presented in Table 7. The final performance of RSPO on the easy map *2m_vs_1z* matches the state-of-the-art (Yu et al., 2021). The median evaluation winning rate and standard deviation on the hard map *2c_vs_64zg* is 98.4%(6.4%), which is slightly lower than the state-of-the-art 100%(0). We note that the policies discovered by our algorithm are *both diverse and high-quality winning strategies (local optima)* showing intelligent emergent behaviors.

We note that population diversity, which we use for quantitative evaluation for *MuJoCo*, may not be an appropriate metric for such a sophisticated MARL game with a large state space and a long horizon. Diversity via Determinant or *Population Diversity* (Parker-Holder et al., 2020b) is originally designed for the continuous control domain. It mainly focuses on the continuous control domain and directly adopts the action as action embedding, while it remains unclear how to embed a discrete action space. For SMAC, we adopt the logarithm probability of categorical distribution as the action embedding and evaluate RSPO and selected baselines using *Population Diversity* on the hard map *2c_vs_64zg*. We further train DvD with a population size of 4 as an additional baseline. The results are shown in Table 8. The population diversity scores of baseline methods and RSPO both reach the maximum value of 1.000. However, if we visualize all the learned policies, actually many policies induced by PBT-CE or DvD cannot be visually distinguished by humans. Moreover, policies induced by PG are visually identical but still achieve a population diversity score of 0.981. This indicates that high population diversity scores might not necessarily imply a diverse strategy pool in complex environments like SMAC. We hypothesize that this is due to complex game dynamics in SMAC. For example, a unit performing micro-strategies of attack-then-move and move-then-attack are visually the same for humans but will have very different action probabilities in each timestep. Such subtle changes in policy outputs can significantly increase the population diversity scores. Since high scores may not directly reflect more diverse policies, it may not be reasonable to explicitly optimize population diversity as an auxiliary loss in SMAC. Therefore, we omit the results of DvD in SMAC in the main body of our paper.

Table 8: *Population Diversity* on the hard map *2c_vs_64zg* in SMAC.

| Algorithm | # Distinct Strategies | Population Diversity |
|---|---|---|
| PG | 1 | 0.981 |
| PBT-CE | 2 | 1.000 |
| DvD | 3 | 1.000 |
| RSPO | 4 | 1.000 |

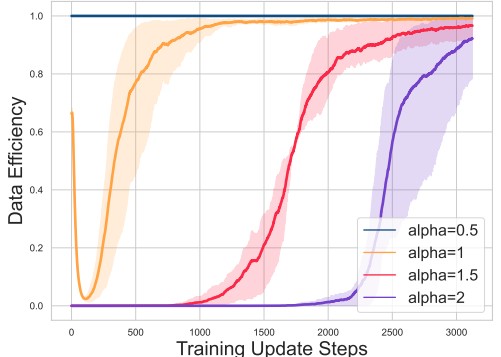 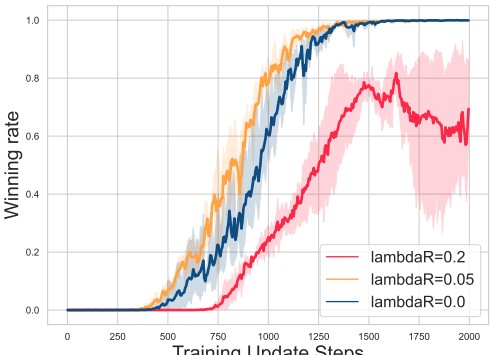

Figure 12: Data efficiency with different $\alpha$ in *Humanoid*.

Figure 13: Learning curve with different $\lambda_{\mathbf{R}}^{int}$ on the *2m_vs_1z* map in SMAC.

| $\alpha$ | *Population Diversity* | $\lambda_{\mathbf{B}}^{int}$ | *Population Diversity* |
|------|------|------|------|
| 0.5 | 0.604 | 0.5 | 0.640 |
| 1.0 | 0.676 | 1.0 | 0.621 |
| 1.5 | 0.687 | 5.0 | 0.697 |
| 2.0 | 0.748 | 10.0 | 0.697 |

Table 9: *Population Diversity* scores of the first 2 policies with different hyperparameters in *Humanoid*. We have scaled the denominator of the RBF kernel in the *Population Diversity* matrix by a factor 10, such that the difference can be demonstrated more clearly.

To the best of our knowledge, a commonly accepted policy diversity metric for complex MARL games remains an open question in the existing literature. In our practice, rendering and visualizing the evaluation trajectories remains the best approach to distinguish different learned strategies. We emphasize that qualitatively, we are so far the first paper that ever reports such a visually diverse collection of winning strategies on a hard map in SMAC. Please check our website for policy visualizations (see Appendix A).

## B.3 SENSITIVITY ANALYSIS

We have performed a sensitivity analysis over $\alpha$, $\lambda_{\mathbf{B}}^{int}$ and $\lambda_{\mathbf{R}}^{int}$ since they are the critical to the performance RSPO. The default values used in our experiments can be found in Table 12.

$\alpha$ is the most important hyperparameter in RSPO because it determines what trajectories in a batch to be accepted. We focus on the *data efficiency*, i.e., the proportion of accepted trajectories in a batch. In the sensitivity analysis, we run the second iteration of RSPO in *Humanoid* with $\alpha = 0.5, 1.0, 1.5, 2$ respectively and compute the population diversity score of the 2 resulting policies. The result is shown in Fig. 12 and the left part of Table 9. The result accords with our heuristic to adjust $\alpha$: with a small $\alpha$ ($\alpha = 0.5$), RSPO may accept all the trajectories at the beginning and lead to a similar policy after convergence, which is no better than the PG baseline; with a large $\alpha$ ($\alpha = 1.5$ and $\alpha = 2$), RSPO may reject too many trajectories at the early stage of training and spend quite a lot of time on exploration, which sacrifices training time for the gain in the diversity score. In practice, we suggest starting with $\alpha = 1$ and adjusting it such that the acceptance rate can drop at the start of training and then quickly and smoothly converge to 1, as shown in Fig. 12 ($\alpha = 1$) and Fig. 5. $\alpha$ should be decreased if too much data is rejected at the beginning of training and increased if data efficiency always stays high in the early stage of training.

$\lambda_{\mathbf{B}}^{int}$ and $\lambda_{\mathbf{R}}^{int}$ determines the scale of intrinsic rewards. In our sensitivity analysis, $\lambda_{\mathbf{B}}^{int}$ is analyzed in the Humanoid environment and $\lambda_{\mathbf{R}}^{int}$ is analyzed on the *2m_vs_1z* map in SMAC. Similarly, we run the second iteration of RSPO with $\lambda_{\mathbf{B}}^{int} = 0.5, 1, 5, 10$ in *Humanoid* and with $\lambda_{\mathbf{R}}^{int} = 0, 0.05, 0.2$ on *2m_vs_1z*. The results are shown in Fig. 13 and the right part of Table 9. With

Table 10: The number of distinct strategies discovered by PGA-MAP-Elites and RSPO and the highest achieved rewards in *4-Goals* Hard. Numbers are averaged over 3 seeds.

|  | Easy | Medium | Hard | *Escalation* | Reward in *4-Goals* Hard |
|---|---|---|---|---|---|
| PGA-MAP-Elites | 4 | 1.6 | 1.2 | 10 | 0.98 |
| RSPO | 4 | 4 | 4 | 7.7 | 1.30 |

value and advantage normalization in PPO, the scale of intrinsic rewards may not significantly affect performance. Specifically, the diversity scores in *Humanoid* do not vary too much, and the induced policies on *2m_vs_1z* are all visually distinct from the reference one. However, if the scale is much larger than extrinsic rewards, it may cause learning instability, as shown in Fig. 13. On the opposite side, if the intrinsic rewards are turned off, RSPO may slow down convergence (Fig. 13), fail to discover non-trivial local optima due to lack of exploration (Table. 1) or get stuck during exploration due to low data efficiency (Fig. 5). We suggest starting with $\lambda_{\mathbf{B}}^{int} = 1$ and $\lambda_{\mathbf{R}}^{int} = 0$, and adjusting them such that the intrinsic rewards lead to a fast and smooth convergence.

### B.4   ADDITIONAL STUDY WITH AN EVOLUTIONARY METHOD

Note that the main paper focuses on the discussion of RL-based solutions which require minimal domain knowledge of the solution structure. Evolutionary methods, as another popular line of research, have also shown promising results in a variety of domains and are also able to discover interesting diverse behaviors (Cully et al., 2015; Hong et al., 2018). However, evolutionary methods typically assume an effective human-designed set of characteristic features for effective learning.

Here, for complete empirical study, we also conduct additional experiments w.r.t. a very recent evolutionary-based algorithm, PGA-MAP-Elites (Nilsson & Cully, 2021) which integrates MAP-Elites (Cully et al., 2015) into a policy gradient algorithm TD3 (Fujimoto et al., 2018). The PGA-MAP-Elites algorithm requires a human-defined behavioral descriptor (BD) to map a neural policy into a low-dimensional (discretized) space for behavior clustering. We run PGA-MAP-Elites on the *4-goals* and the *Escalation* environment, where the behavioral descriptors (BDs) can be precisely defined to the best of our efforts. We remark that for the remaining cases, including the *Monster-Hunt* environment, MuJoCo control domain, and SMAC scenarios, behaviors of interests always involve strong *temporal characteristics*, which makes the design of a good BD particularly non-trivial and remain a challenging open question for the existing literature.

In particular, we define a 4-dimensional descriptor for the *4-Goals* environment, i.e., a one-hot vector indicating the ID of the nearest landmark. For the *Escalation* environment, we use a 1-dimensional descriptor for the *Escalation* environment, which is a 0-1-normalized value of the cooperation steps within the episode. We set the number of behavior cells (niches) equal to the iteration number in RSPO, specifically 7 for *4-Goals* and 10 for *Escalation*.

Results are shown in Table 10, where PGA-MAP-Elites performs much worse on the *4-Goals* scenario while outperforms RSPO on the *Escalation* environment due to the informative BD. Based on the results, we would like to discuss some characteristics of evolutionary algorithms and RSPO below:

1. *We empirically observe that many policies produced by PGA-MAP-Elites are immediately archived without becoming converged*, particularly in *4-Goals* (Hard) and *Escalation*. When measuring population diversity, these unconverged policies would contribute a lot even though many of them may have unsatisfying behaviors/returns. By contrast, the objective of RSPO aims to find diverse *local optima*. This also suggests a further research direction to bridge such a convergence-diversity gap.

2. *The quality of BD can strongly influence the performance of evolutionary methods.* Note that the BD in *Escalation* provides a particularly clear signal on whether a policy reaches a local optimum or not (i.e., each BD niche precisely corresponds to a policy mode) while RSPO directly works on the deceptive extrinsic reward structure without knowing the structure of NEs. This suggests the importance of BD design, which, however, remains an open challenge in general.

3. *An improper BD may lead to a largely constrained behavior space.* The success of PGA-MAP-Elites largely depends on the fact that the BD is known to be able to effectively cover all the local optima of interest. However, for complex environments like SMAC, we do not even know in advance what kind of behaviors will emerge after training. Therefore, an open-ended BD would be desired, which becomes an even more challenging problem — note that there even has not been any effective diversity measurement on SMAC yet. Therefore, a purely RL-based solution would be preferred.

4. *Without an informative BD, evolutionary methods typically require a large population size introduced, which can cause practical issues.* For example, maintaining a large unstructured archive can be computationally expensive. It can be also challenging to visually evaluate learned behaviors given a large population.

To sum up, when an effective and informative BD is available, evolutionary methods can be a strong candidate to consider, although it may not fit every scenario of interest, while RSPO can be a generally applicable solution with minimal domain knowledge. It could be also beneficial to investigate how to incorporate informative domain prior into RSPO framework, which we leave as our future work.

### B.5 THE POINT-V0 ENVIRONMENT

Parker-Holder et al. (2020b) develops a continuous control environment which requires the agent to bypass a wall to reach the goal. A big penalty will be given to the agent if it directly runs towards the goal and hits the wall. This environment indeed has a local optimum which can be overcome by diversity-driven exploration. While the authors of the DvD paper argued that ES and NSR will get stuck in the wall, the naive PPO algorithm can directly learns to bypass the wall and escape the local optimum. Hence in our experiment section, we consider much more challenging environments where a much larger number of local optima exist, such as stag-hunt games and SMAC.

### B.6 SMERL

In SMERL (Kumar et al., 2020), if the agent can achieve sufficiently high return in the trajectory, the trajectory will be augmented with intrinsic rewards of DIAYN (Eysenbach et al., 2019) for skill differentiation and policy diversification. In *Humanoid*, it may be challenging for SMERL to achieve such a high return, which turns off the intrinsic reward for promoting diversity. Hence, we do not report the population diversity score in the main body of our paper.

In stag-hunt games, SMERL keeps producing low returns. Note that since the SMERL algorithm only starts promoting diversity after sufficiently high reward is achieved, all the produced policies by SMERL are visually identical non-cooperative strategies. What's more, DIAYN (Eysenbach et al., 2019), which has the same intrinsic reward as SMERL, has been evaluated in Tang et al. (2021) and proved to perform worse than RPG (Tang et al., 2021), while the performance of RPG is further surpassed by RSPO. Hence, we omit the results of SMERL in the stag-hunt games.

We also evaluate SMERL in SMAC with a latent dimension of 5, which only induces 1 strategy on each map across all possible latent variables. We hypothesize the reason is that in such a complex MARL environment with a large state space and long horizon, the skill latent variable may be particularly challenging to be determined from game states. Moreover, the latent dimension is usually dominated by the state dimension, which makes latent variables less effective.

## C ENVIRONMENT DETAILS

### C.1 4-GOALS

The implementation of *4-Goals* is based on *Multi-Agent Particle Environments*[1] (Mordatch & Abbeel, 2018). The size of the agent is 0.02 and the size of the goals are 0.04 (in easy and medium mode). The agent spawns at $(0, 0)$. In the easy mode, the four goals spawn at $\{(0, 0.5), (0.5, 0), (0, -0.5), (-0.5, 0)\}$ respectively; In the medium and hard mode, the four goals spawn uniformly randomly with $x \in [-1, 1]$ and $y \in [-1, 1]$. At each step, the agent receives an

---

[1]https://github.com/openai/multiagent-particle-envs

observation vector of the relative positions of the four goals to itself. The game ends when the agent touches any of the goals or when the timestep limit of 16 is reached.

## C.2   GRIDWORLD STAG-HUNT GAMES

Both *Monster-Hunt* and *Escalation* are based on a $5 \times 5$ grid. In both games, the action space contains 5 discrete actions, including moving up, moving down, moving up, moving down and stay. The policy output is a softmax distribution over these actions. Each movement action moves the agent to one neighboring grid while the stay action makes the agent stay still. Each episode lasts for 50 timesteps. We used vector representations with continuous values for observations, i.e., two real numbers to represent the 2-dimensional positions for each entity. The detailed state representations are presented in the following sub-sections respectively. (Note: the dimension of each feature is shown in the parentheses)

### C.2.1   MONSTER-HUNT

*Monster-Hunt* is based on a $5 \times 5$ grid. There are three types of entities: 2 agents, 1 monster and 2 apples. All of the entities spawn uniformly randomly on the grid without overlapping. At each step, both agents can choose to move up, down, left, right or stay; The monster always moves for one step towards its closest agent (ties break randomly). If one agent ends up in the same grid with the monster, it receives a penalty of -2; If both agents end up in the same grid with the monster, they both receive of bonus of 5; If any agent ends up in the same grid with an apple, it receives a bonus of 1 (ties break randomly if both agents enter the apple grid at the same time). Whenever any agent ends up in the same grid with a non-agent entity, the entity respawns randomly on the grid. The observation vector of an agent consists of the absolute position of itself and the relative positions of the other entities. Specifically, total dimension is 10, consisting of self absolute position (2), other agent's relative position (2), monster's relative position (2), apple 1's relative position (2), apple 2's relative position (2).

### C.2.2   ESCALATION

Same as *Monster-Hunt*, *Escalation* is also based on a $5 \times 5$ grid. There are two types of entities: 2 agents and 1 light. All of the entities spawn uniformly randomly on the grid without overlapping. When the two agents and the light ends up in the same grid for the first time, they enter *cooperation* stage and the cooperation length $L$ is set to 0. At each step of *cooperation* stage, they first both receive a bonus of 1, then $L$ increases by 1, the light entity moves to a random adjacent grid and the agents move again. The cooperation stage continues only if both agents move the the same grid as the light. If only one agent moves to the light, it receives a penalty of $-0.9L$. The game ends if the cooperation stage ends. The observation vector of an agent consists of the current cooperation length $L$, the absolute position of itself and the relative positions of the other entities. Specifically, total dimension is 7, consisting of current cooperation length (1), self absolute position (2), other agent's relative position (2), light's relative position (2).

## C.3   MUJOCO

We use the MuJoCo environments from Gym (version 0.17.3) with a moving forward reward. The only modification is that we set the episode length to be 512 so that the diverse intrinsic rewards can be easily computed.

## C.4   STARCRAFT MULTI-AGENT CHALLENGE

SMAC concentrates on decentralized micromanagement scenarios, where each unit of the game is controlled by an individual RL agent. Detailed description of state/action space and reward function can be found in Rashid et al. (2019).

Table 11: PPO hyperparameters for different experiments.

| Hyperparameters | 4-Goals | Stag-Hunt Games | MuJoCo | SMAC |
|---|---|---|---|---|
| Network structure | MLP | MLP | MLP | MLP+RNN[1] |
| Hidden size | 64 | 64 | 64 | 64 |
| Initial learning rate | 3e-4 | 1e-3 | 3e-4 | 5e-4 |
| Batch size | 8192 | *Monster-Hunt*: 12800 *Escalation*: 6400 | 16384 | 1600[2] |
| Minibatch size | 8192 | *Monster-Hunt*: 12800 *Escalation*: 6400 | 512 | 1600[2] |
| Adam stepsize ($\epsilon$) | 1e-5 | 1e-5 | 1e-5 | 1e-5 |
| Discount rate ($\gamma$) | 0.99 | 0.99 | 0.99 | 0.99 |
| GAE parameter ($\lambda$) | 0.95 | 0.95 | 0.95 | 0.95 |
| Value loss coefficient | 0.5 | 1.0 | 0.5 | 0.5 |
| Entropy coefficient | 0.05 | 0.01 | 0.01 | 0.01 |
| Gradient clipping | 0.5 | 0.5 | 0.5 | 10.0 |
| PPO clipping parameter | 0.2 | 0.2 | 0.2 | 0.2 |
| PPO epochs | 4 | 4 | 10 | *2m_vs_1z*: 1 *2c_vs_64zg*: 5 |

[1] Gated Recurrent Unit (GRU) (Chung et al., 2014)
[2] 1600 chunks with length 10.

## D  IMPLEMENTATION DETAILS

For all experiments, baselines are re-initialized and re-run multiple times with the same number of iterations as RSPO except for population-based methods. Population-based baselines are run for a single trial with a population of policies trained in parallel. We also make sure the population size is the same as the iteration number of RSPO. All the baselines are built upon PPO with the same hyperparameters and run for the same number of total environment steps.

### D.1  RSPO

We use PPO with separate policy and value networks as the algorithm backbone of RSPO in all experiments. Both policy and value networks are multi-layer perceptrons (MLP) with a single hidden layer. Specifically for SMAC, we add an additional GRU (Chung et al., 2014) layer with the same hidden size and adopt the paradigm of centralized-training-with-decentralized-execution (CTDE), under which agents share a policy conditioned on local observations and a value function conditioned on global states. The PPO hyperparameters we use for each experiment is shown in Table 11. Even though the trajectory filtering technique induces an unbiased policy gradient estimator, it introduces unnegligible variance in the early stage of training. Therefore, we typically use a large batch size for PPO training to alleviate the defect of trajectory filtering. What's more, by using a large batch size, the advantage normalization and value normalization tricks could empirically improve learning stability. In the late stage of training, more trajectories are accepted and learning smoothly converges to a locally optimal solution as shown in Fig .5. We also note that the hidden size is 64 across the 4 domains, which may hurt final performance especially in the *MuJoCo* environments.

There are three additional hyperparameters for RSPO: the automatic threshold coefficient $\alpha$ mentioned in Section 3.5, weight of the behavior-driven intrinsic reward $\lambda_{\mathbf{B}}^{int}$ and weight of the reward-driven intrinsic reward $\lambda_{\mathbf{R}}^{int}$. These hyperparameters are shown in Table 12.

**Smoothed-switching**  We mention the *smoothed-switching* technique in Section 3.5 and remark that it helps to stabilize training when there are a large number of reference policies. Therefore, we apply *smoothed-switching* in experiments with 10+ iterations, specifically in Fig. 4 and Fig. 7b. We use the average of switching indicators in a batch as the smoothed indicator.

Table 12: RSPO hyperparameters for different experiments.

| Hyperparameters | *4-Goals* | *Stag-Hunt Games* | *MuJoCo* | *SMAC* |
|:---:|:---:|:---:|:---:|:---:|
| $\lambda_{\mathbf{B}}^{int}$ | *N/A* | 0.2 | 1 | *2m_vs_1z*: 1
*2c_vs_64zg*: 5 |
| $\lambda_{\mathbf{R}}^{int}$ | *N/A* | 1 | 0 | *2m_vs_1z*: 0.05
*2c_vs_64zg*: 0 |
| $\alpha$ | 0.5 | 0.6 | *H.Cheetah*: 1.1
*Hopper* & *Walker2d*: 0.9
*Homanoid*: 1.0 | *2m_vs_1z*: 0.5
*2c_vs_64zg*: 0.3 |

## D.2 BASELINE METHODS

All baseline methods are integrated with PPO and utilize the same hyperparameters as presented in Table 11. For population-based methods, all workers have independent data collection processes and buffers. RSPO and all baseline methods except for PG are initialized with the same weights. If not specially mentioned, the algorithm-specific hyperparameters are inherited from other papers which proposed the algorithm or utilized the algorithm as a baseline.

**PG**  Policy gradient with restarts (PG) utilize different random seeds for different policies. Samples are collected and consumed independently during training.

**PBT-CE**  Population-based training with our cross entropy objective (PBT-CE) utilizes the intrinsic reward in Eq. 3 as a soft objective without trajectory filtering and reward switching. By comparing PBT-CE with RSPO, we can validate the significance of the iterative learning and reward-switching scheme. We performed a grid search over 1e0, 5e-1, 1e-2, 5e-2, 5e-3, 2e-3, and 1e-3 on the Lagrange multiplier in each domain, and only 1e-3 converges.

**DIPG**  DIPG (Masood & Doshi-Velez, 2019) utilizes the Maximum Mean Discrepancy (MMD) of the *state distribution* between the learning policy and reference policies as a diversity driven objective. By contrast, our algorithm mainly focuses on the divergence of *action distribution* and optionally augment with approximate divergence of state distribution. We performed a grid search over 2e0, 1e0, 5e-1, 1e-1, and 1e-2 on the coefficient of MMD loss in each domain. The MMD loss coefficient is fixed to 0.1 in SMAC and 1.0 in other environments. We refer to the public implementation[2].

**MAVEN**  We use the open-source implementation[3].

**PGA-MAP-Elites**  We use the open-source implementation[4].

**DvD**  Diversity via Determinant (DvD) (Parker-Holder et al., 2020b) algorithm aims to maximize the determinant of the RBF kernel matrix of action embedding. The original paper mainly focuses on the continuous control domain and directly adopts the action as action embedding, while it remains unclear how to embed a discrete action space. Regarding implementation, we refer to the public codebase[5]. We also utilize the Bayesian Bandits module to automatically adjust the coefficient of the auxiliary loss from $\{0, 0.5\}$, same as the original paper.

**SMERL**  In SMERL (Kumar et al., 2020), if the agent can achieve sufficiently high return in the trajectory, the trajectory will be augmented with intrinsic rewards of DIAYN (Eysenbach et al., 2019) for skill differentiation. SMERL is mainly designed for tackling perturbation in the continuous control domain. We use the score obtained in the first iteration of RSPO as the expert return for SMERL. We train SMERL with a latent dimension of 5 with intrinsic reward coefficient $\alpha = 10.0$

---

[2]https://github.com/dtak/DIPG-public
[3]https://github.com/AnujMahajanOxf/MAVEN
[4]https://github.com/ollenilsson19/PGA-MAP-Elites
[5]https://github.com/jparkerholder/DvD_ES

and trajectory threshold raio $\epsilon = 0.1$, same as the original paper. For a fair comparison, we train SMERL for latent-dim$\times$ more environment frames.

**TrajDiv**    Trajectory Diversity (TrajDiv) (Lupu et al., 2021) is designed for tackling zero-shot coordination problem in cooperative multi-agent games, specifically Hanabi (Bard et al., 2020). Trajdiv utilizes the approximate Jensen-Shannon divergence with action discounting kernel as an auxiliary loss in population-based training. We performed a grid search over 1e-2, 1e-1, 1e0 on the coefficient on the TrajDiv loss and over 1e-1, 5e-1, 9e-1 on the action-kernel discounting factor. We set the coefficient of trajectory diversity loss to be $0.1$ and action discount factor to be $0.9$.

**Ridge Rider**    The Ridge Rider method (Parker-Holder et al., 2020a) attempts to search for local optima via Hessian information. However, due to the high variance of policy gradients, the Hessian information can not be accurately estimated in complex RL problems. The Ridge Rider method basically diverges across the 4 domains we considered. Hence, we exclude the results of it.

## E    POPULATION DIVERSITY

*Population Diversity (PD)* is originally proposed in Parker-Holder et al. (2020b) as a metric to measure the group diversity of a population of deterministic policies $\Pi = \{\mu_1, \ldots, \mu_M\}$. The original PD is defined as:

$$\text{Div}_{\text{DvD}}(\Pi) := \det(K_{\text{SE}}(\phi(\mu_i), \phi(\mu_j))_{i,j=1}^M) \tag{10}$$

where $\phi$ is a *Behavior Embedding* of $\pi$ defined as a concatenation of the actions taken on $N$ randomly sampled states and $K : \mathbb{R}^d \times \mathbb{R}^d \to \mathbb{R}$ is a kernel function. In the case of Parker-Holder et al. (2020b), $K$ is the Squared Exponential (or RBF) kernel, defined as $K_{\text{SE}}(x, y) := \exp\left(-\frac{\|x-y\|^2}{2l^2N}\right)$ where $l$ is a *length scale*.

In Parker-Holder et al. (2020b), only deterministic policies were considered. In our work, we design an modified version of PD that suits a set of stochastic policies $\Pi = \{\pi_1, \ldots, \pi_M\}$, defined as follows:

$$\text{Div}(\Pi) := \det(K_{\text{JSD}}(\pi_i, \pi_j)_{i,j=1}^M) \tag{11}$$

where $K_{\text{JSD}}(x, y) := \exp\left(-\frac{JSD(x\|y)}{2p^2}\right)$ and $JSD$ is the Jensen-Shannon divergence. We provide the following theorem that connects our definition of PD with the original PD in continuous action space.

**Theorem E.1.** *Let $\Pi = \{\mu_1, \ldots, \mu_M\}$ be a set of deterministic policies in continuous action space. Let $\hat{\Pi} = \{\pi_1, \ldots, \pi_M\}$ be the randomized policies of $\Pi$ defined as $\pi_i(a|s) \sim \mathcal{N}(\mu_i(s), \sigma^2)$ where $\sigma > 0$ is an arbitrary deviation. With a suitable choice of p, $\text{Div}(\hat{\Pi}) = \lim_{N\to\infty} \text{Div}_{DvD}(\Pi)$ if the states in $\phi$ is sampled with the randomized policies.*

*Proof.* We first observe that for any $i, j \in \{1, \ldots, M\}$,

$$\lim_{N\to\infty} \frac{\|\phi(\mu_i) - \phi(\mu_j)\|^2}{N} = \lim_{N\to\infty} \frac{1}{N} \sum_{k=1}^N \|\mu_i(s_k) - \mu_j(s_k)\|^2$$
$$= \mathbb{E}_{s\sim\mu_i\cup\mu_j} \left[\|\pi_i(s) - \pi_j(s)\|^2\right] \tag{12}$$

In the case of stochastic policies, we have

$$
\begin{aligned}
D_{KL}(\pi_i \| \pi_j) =& \mathbb{E}_{s,a \sim \pi_i} \left[ \log \frac{\pi_i(a|s)}{\pi_j(a|s)} \right] \\
=& \mathbb{E}_{s,a \sim \pi_i} \left[ -\frac{1}{2\sigma^2} \left( \|a - \mu_i(s)\|^2 - \|a - \mu_j(s)\|^2 \right) \right] \quad \text{(Since } \pi_i(a|s) \sim \mathcal{N}(\mu(s), \sigma^2)) \\
=& \frac{1}{2\sigma^2} \mathbb{E}_{s \sim \pi_i} \left[ (\|\mu_i(s) - \mu_j(s)\|^2 + \sigma^2) - \sigma^2 \right] \quad \text{(Since } a \sim \pi_i(s)) \\
=& \frac{1}{2\sigma^2} \mathbb{E}_{s \sim \pi_i} \left[ \|\mu_i(s) - \mu_j(s)\|^2 \right]
\end{aligned}
\tag{13}
$$

Then,

$$
\begin{aligned}
JSD(\pi_i \| \pi_j) =& \frac{1}{2} D_{KL}(\pi_i, \pi_j) + \frac{1}{2} D_{KL}(\pi_j, \pi_i) \\
=& \frac{1}{4\sigma^2} \mathbb{E}_{s \sim \pi_i} \left[ \|\mu_i(s) - \mu_j(s)\|^2 \right] + \frac{1}{4\sigma^2} \mathbb{E}_{s \sim \pi_j} \left[ \|\mu_j(s) - \mu_i(s)\|^2 \right] \\
=& \frac{1}{2\sigma^2} \mathbb{E}_{s \sim \pi_i \cup \pi_j} \left[ \|\mu_i(s) - \mu_j(s)\|^2 \right]
\end{aligned}
\tag{14}
$$

Therefore, if we choose $p = \frac{l}{\sigma}$,

$$
\begin{aligned}
\text{Div}(\hat{\Pi}) =& \det(K_{\text{JSD}}(\pi_i, \pi_j)_{i,j=1}^M) \\
=& \det \left( \exp \left( -\frac{JSD(\pi_i \| \pi_j)}{2p^2} \right)_{i,j=1}^M \right) \\
=& \det \left( \exp \left( -\frac{\sigma^2 JSD(\pi_i \| \pi_j)}{2l^2} \right)_{i,j=1}^M \right) \\
=& \det \left( \exp \left( \lim_{N \to \infty} -\frac{\|\phi(\mu_i) - \phi(\mu_j)\|^2}{2l^2 N} \right)_{i,j=1}^M \right) \\
=& \lim_{N \to \infty} \det \left( \exp \left( -\frac{\|\phi(\mu_i) - \phi(\mu_j)\|^2}{2l^2 N} \right)_{i,j=1}^M \right) \\
=& \lim_{N \to \infty} \text{Div}_{\text{DvD}}(\Pi)
\end{aligned}
\tag{15}
$$

$\square$

## F   MOTIVATION OF USING CROSS-ENTROPY DIVERSITY MEASURE

Maximizing KL-divergence between policies could be problematic in RL problems. The accumulative KL-divergence of a trajectory is defined as

$$
\begin{aligned}
D_{KL}(\pi_i, \pi_j) =& \mathbb{E}_{\tau \sim \pi_i} \left[ \sum_t \log \frac{\pi_i(a_t|s_t)}{\pi_j(a_t|s_t)} \right] \\
=& \underbrace{\mathbb{E}_{\tau \sim \pi_i} \left[ -\sum_t \log \pi_j(a_t|s_t) \right]}_{\text{cross entropy}} - \underbrace{\mathbb{E}_{\tau \sim \pi_i} \left[ -\sum_t \log \pi_i(a_t|s_t) \right]}_{\text{entropy of } \pi_i}.
\end{aligned}
$$

The above derivation suggests that optimizing KL-divergence would inherently encourages learning a policy with small entropy. This is usually undesirable. Therefore, we choose to use cross-entropy instead of KL-divergence.

## G    DERIVATION OF TRAJECTORY FILTERING AND REWARD-SWITCHING

In this section, we want to validate our claim that solving the trajectory filtering objective in Eq. (5) and the reward switching objective in Eq. (6) is equivalent to solving Eq. (2) with an even stronger diversity measure $D_{\text{filter}}(\pi_i, \pi_j)$ defined by

$$D_{\text{filter}}(\pi_i, \pi_j) = \inf_{\tau \sim \pi_i} \left\{ -\sum_t \log \pi_j (a_t | s_t) \right\}.$$

Note that $D_{\text{filter}}(\pi_i, \pi_j)$ measures *lowest* possible trajectory-wise negative likelihood rather than the *average* value (i.e., cross-entropy) in Eq. (3).

For simplicity, in the following discussions, we omit the discounted factor $\gamma$ for conciseness. Before presenting the theoretical justifications, we focus on a simplified setting with the following assumptions.

**Assumption 1.** *(Non-negative Reward and Fixed Horizon) The MDP has finite states and actions with a fixed horizon length and non-negative rewards, i.e., $r_t \geq 0$ at each timestep $t$.*

We remark that this is a general condition since for any fixed-horizon MDP with bounded rewards, we can shape the reward function without changing the optimal policies by adding a big constant to each possible reward.

**Assumption 2.** *(Multiple Distinct Global Optima) There exist $M$ distinct global optimal policies $\pi_1^*, \pi_2^*, \dots, \pi_M^*$, namely, $\pi_i^* \in \arg\max_\pi \mathbb{E}_{\tau \sim \pi} [\sum_t r_t]$ for $1 \leq i \leq M$, and $D_{\text{filter}}(\pi_i^*, \pi_j^*) \geq \delta$ for $1 \leq i \neq j \leq M$.*

This assumption ensures that the problem is solvable.

**Assumption 3.** *(Non-trivial Optimum) For each global optimal policy $\pi^*$, we have $\sum_{r_t \in \tau} r_t > 0$ for each possible trajectory $\tau$ under the policy $\pi^\star$.*

Since rewards are non-negative, Assumption 3 suggests that the optimal policy will not generate any trivial trajectories. We also remark that similar to Assumption 1, this assumption can be generally satisfied by properly shaping the underlying reward function without changing the optimal policies.

With all the assumptions presented, we first derive the filtering objective in Eq. (5) via Theorem G.1 and then the switching objective in Eq. (6) via Theorem G.2.

**Theorem G.1.** *(Filtering Objective) Consider the constrained optimization problem*

$$\pi_{k+1} = \arg\max_\pi \mathbb{E}_{\tau \sim \pi} \left[ \sum_t r_t \right], \;\; \text{subject to} \;\; \inf_{\tau \sim \pi} \left\{ -\sum_t \log \pi_i (a_t | s_t) \right\} \geq \delta, \; \forall 1 \leq i \leq k < M. \tag{16}$$

*Given assumption 1, 2 and 3, solving the following unconstrained optimization problem*

$$\hat{\pi}_{k+1} = \arg\max_\pi \mathbb{E}_{\tau \sim \pi} \left[ \Phi_k(\tau) \sum_t r_t \right] \tag{17}$$

*is equivalent to solving Eq. (16), where $\Phi_k(\tau) := \prod_{i=1}^k \phi_i(\tau)$ and $\phi_i(\tau)$ is defined by*

$$\phi_i(\tau) = \begin{cases} 1, & \text{if } -\sum_t \log \pi_i(a_t|s_t) \geq \delta, \; (s_t, a_t) \sim \tau \\ 0, & \text{otherwise} \end{cases}. \tag{18}$$

*Proof.* When $k = 1$, the optimization problems in Eq. (16) and Eq. (17) are trivially equivalent. We assume the optimization in the first iteration leads to a global optimum. For any $k > 1$, given previously discovered optimal policies $\{\pi_i | 1 \leq i \leq k\}$, we will show that Eq. (17) must have an equivalent solution to Eq. (16), which is a new global optimum. By induction, the proposition holds.

Consider a fixed $k$, we use $\mathcal{S}$ to denote the subspace of feasible policies satisfying all the distance constraints, i.e., $\mathcal{S} = \{\pi \,|\, \forall 1 \leq i \leq k, \; D_{\text{filter}}(\pi, \pi_i) \geq \delta\}$. Note that by Assumption 2, We would like to prove Theorem. G.1 by showing that the optimal value of Eq. (17) over $\pi \in \mathcal{S}$ is greater than the value over $\pi \notin \mathcal{S}$. Consequently, the solution of Eq. (17) should satisfy the constraints in Eq. (16).

For $\pi \in \mathcal{S}$,

$$\max_{\pi \in \mathcal{S}} \mathbb{E}_{\tau \sim \pi}[\Phi_k(\tau) \textstyle\sum_t r_t] = \max_{\pi \in \mathcal{S}} \mathbb{E}_{\tau \sim \pi} \left[ \sum_t r_t \right] = \max_{\pi \; \mathbb{E}_{\tau \sim \pi}[\sum_t r_t]}. \tag{19}$$

For $\pi \notin \mathcal{S}$, we define

$$T = \left\{ \tau \sim \pi \;\middle|\; -\sum_t \log \pi_i\left(a_t | s_t\right) \geq \delta \;,\; \forall 1 \leq i \leq k \right\} \tag{20}$$

as the constraint-satisfying set of $\tau$ under policy $\pi$. The right-hand side of Eq. (17) can be written as

$$\max_{\pi \notin \mathcal{S}} \text{RHS} = \max_{\pi \notin \mathcal{S}} \mathbb{E}_{\tau \sim \pi} \left[ \Phi_k(\tau) \sum_t r_t \right]$$

$$= \max_{\pi \notin \mathcal{S}} \mathbb{P}_\pi(\tau \in T) \mathbb{E}_{\tau \sim \pi} \left[ \Phi_k(\tau) \sum_t r_t \;\middle|\; \tau \in T \right] + \mathbb{P}_\pi(\tau \notin T) \mathbb{E}_{\tau \sim \pi} \left[ \Phi_k(\tau) \sum_t r_t \;\middle|\; \tau \notin T \right]$$

$$= \max_{\pi \notin \mathcal{S}} \mathbb{P}_\pi(\tau \in T) \mathbb{E}_{\tau \sim \pi} \left[ 1 \cdot \sum_t r_t \;\middle|\; \tau \in T \right] + \mathbb{P}_\pi(\tau \notin T) \mathbb{E}_{\tau \sim \pi} \left[ 0 \cdot \sum_t r_t \;\middle|\; \tau \notin T \right]$$

$$= \max_{\pi \notin \mathcal{S}} \mathbb{P}_\pi(\tau \in T) \mathbb{E}_{\tau \sim \pi} \left[ \sum_t r_t \;\middle|\; \tau \in T \right]. \tag{21}$$

Here $\mathbb{E}_{\tau \sim \pi}\left[\cdot \,|\, \tau \in T\right]$ denotes the expectation conditioned on the event $\{\tau \in T\}$ and $\mathbb{P}_\pi(\tau \in T) = \mathbb{E}_{\tau \sim \pi}\left[\Phi_k(\tau)\right]$.

Next, we want to prove that

$$\max_{\pi \notin \mathcal{S}} \text{RHS of Eq. (17)} < \max_{\pi \notin \mathcal{S}} \mathbb{E}_{\tau \sim \pi} \left[ \sum_t r_t \right]. \tag{22}$$

On one side, if the optimal value of the right-hand side in Eq. (17) is not a global optimum, Eq. (22) holds because $r_t \geq 0$ by Assumption 1. On the other side, we consider the solution of the right-hand side in Eq. (17) is a globally optimal policy $\pi^* \notin \mathcal{S}$. According to the definition of $T$, $\mathbb{P}_{\pi^*}(\tau \notin T) > 0$. By Assumption 3, for $\forall \tau \sim \pi^*$, $\sum_\tau r_t > 0$. Further, for $\pi^* \notin \mathcal{S}$, $\mathbb{P}_\pi(\tau \notin T) \mathbb{E}_{\tau \sim \pi}\left[\sum_t r_t \,|\, \tau \notin T\right] > 0$. Thus,

$$\max_{\pi \notin \mathcal{S}} \text{RHS of Eq. (17)} < \max_{\pi \notin \mathcal{S}} \mathbb{P}_\pi(\tau \in T) \mathbb{E}_{\tau \sim \pi} \left[ \sum_t r_t \;\middle|\; \tau \in T \right] + \mathbb{P}_\pi(\tau \notin T) \mathbb{E}_{\tau \sim \pi} \left[ \sum_t r_t \;\middle|\; \tau \notin T \right]$$

$$= \max_{\pi \notin \mathcal{S}} \mathbb{E}_{\tau \sim \pi} \left[ \sum_t r_t \right] = \max_{\pi} \mathbb{E}_{\tau \sim \pi} \left[ \sum_t r_t \right] = \max_{\pi \in \mathcal{S}} \mathbb{E}_{\tau \sim \pi} \left[ \sum_t r_t \right]. \tag{23}$$

Therefore, we can conclude that the maximum value of the objective function in Eq. (17) should be obtained when $\pi \in \mathcal{S}$, because for $\pi \in \mathcal{S}$, $\phi_i(\tau) = 1$ for all $1 \leq i \leq k$, the constrained objective and the unconstrained objective have the same form. Finally, by induction, the solution of Eq. (17) must be a solution of Eq. (16). $\square$

**Theorem G.2.** *(Switching Objective) Consider the constrained optimization problem in Eq. (16). Given Assumption. 1, 2 and 3, for any $\delta > 0$, there exists some $\lambda > 0$ such that solving the following unconstrained optimization problem*

$$\hat{\pi}_{k+1} = \arg \max_\pi \mathbb{E}_{\tau \sim \pi} \left[ \Phi_k(\tau) \sum_t r_t + \lambda \sum_{i=1}^k (1 - \phi_i(\tau)) \left( -\sum_t \log \pi_i(a_t | s_t) \right) \right] \tag{24}$$

*is equivalent to solving Eq. (16), where $\Phi_k(\tau) := \prod_{i=1}^k \phi_i(\tau)$ and $\phi_i(\tau)$ is defined by*

$$\phi_i(\tau) = \begin{cases} 1, & if \;\; -\sum_t \log \pi_i(a_t | s_t) \geq \delta, \; (s_t, a_t) \sim \tau \\ 0, & otherwise \end{cases}. \tag{25}$$

*Proof.* Following the same induction process and definition of $\mathcal{S}$, the critical part is again to show that the optimal value of Eq. (24) w.r.t. a particular iteration $k$ over $\pi \in \mathcal{S}$ is greater than the value over $\pi \notin \mathcal{S}$. Consequently, the solution of Eq. (24) should satisfy the constraints in Eq. (16) and the overall proposition holds by induction.

For $\pi \in \mathcal{S}$, the unconstrained optimization problem in Eq. (24) changes to

$$\hat{\pi}_{k+1} = \arg\max_{\pi \in \mathcal{S}} \mathbb{E}_{\tau \sim \pi} \left[ \sum_t r_t \right]. \tag{26}$$

For $\pi \notin \mathcal{S}$, we define

$$T = \left\{ \tau \sim \pi \,\middle|\, -\sum_t \log \pi_i (a_t | s_t) \geq \delta , \; \forall 1 \leq i \leq k \right\} \tag{27}$$

as the constraint-satisfying set of $\tau$ given a policy $\pi$. First we observe that when $\tau \notin T$,

$$\sum_{i=1}^{k} (1 - \phi_i(\tau)) \left( -\sum_t \log \pi_i(a_t|s_t) \right)$$

$$= \sum_{i=1}^{k} \mathbb{1} \left[ -\sum_t \log \pi_i(a_t|s_t) < \delta \right] \left( -\sum_t \log \pi_i(a_t|s_t) \right)$$

$$< \sum_{i=1}^{k} \delta = k\delta. \tag{28}$$

Then we can write the right-hand side of Eq. (24) as:

$$\max_{\pi \notin \mathcal{S}} \text{RHS} = \max_{\pi \notin \mathcal{S}} \mathbb{E}_{\tau \sim \pi} [g(\tau)]$$

$$= \max_{\pi \notin \mathcal{S}} \mathbb{P}_\pi(\tau \in T) \mathbb{E}_{\tau \sim \pi} [g(\tau) | \tau \in T] + \mathbb{P}_\pi(\tau \notin T) \mathbb{E}_{\tau \sim \pi} [g(\tau) | \tau \notin T]$$

$$= \max_{\pi \notin \mathcal{S}} \mathbb{P}_\pi(\tau \in T) \mathbb{E}_{\tau \sim \pi} \left[ \Phi_k(\tau) \sum_t r_t + \lambda \sum_{i=1}^{k} (1 - \phi_i(\tau)) \left( -\sum_t \log \pi_i(a_t|s_t) \right) \,\middle|\, \tau \in T \right]$$

$$+ \mathbb{P}_\pi(\tau \notin T) \mathbb{E}_{\tau \sim \pi} \left[ \Phi_k(\tau) \sum_t r_t + \lambda \sum_{i=1}^{k} (1 - \phi_i(\tau)) \left( -\sum_t \log \pi_i(a_t|s_t) \right) \,\middle|\, \tau \notin T \right]$$

$$= \max_{\pi \notin \mathcal{S}} \mathbb{P}_\pi(\tau \in T) \mathbb{E}_{\tau \sim \pi} \left[ 1 \cdot \sum_t r_t \,\middle|\, \tau \in T \right] + \mathbb{P}_\pi(\tau \notin T) \mathbb{E}_{\tau \sim \pi} \left[ 0 \cdot \sum_t r_t \,\middle|\, \tau \notin T \right]$$

$$+ \lambda \mathbb{P}_\pi(\tau \notin T) \mathbb{E}_{\tau \sim \pi} \left[ \sum_{i=1}^{k} (1 - \phi_i(\tau)) \left( -\sum_t \log \pi_i(a_t|s_t) \right) \,\middle|\, \tau \notin T \right]$$

$$< \max_{\pi \notin \mathcal{S}} \mathbb{P}_\pi(\tau \in T) \mathbb{E}_{\tau \sim \pi} \left[ 1 \cdot \sum_t r_t \,\middle|\, \tau \in T \right] + \mathbb{P}_\pi(\tau \notin T) \mathbb{E}_{\tau \sim \pi} \left[ 0 \cdot \sum_t r_t \,\middle|\, \tau \notin T \right] + \lambda k\delta$$

$$= \max_{\pi \notin \mathcal{S}} \mathbb{E}_{\tau \sim \pi} \left[ \sum_t r_t \right] - \mathbb{P}_\pi(\tau \notin T) \mathbb{E}_{\tau \sim \pi} \left[ \sum_t r_t \,\middle|\, \tau \notin T \right] + \lambda k\delta. \tag{29}$$

Here $\mathbb{E}_{\tau \sim \pi} [\cdot | \tau \in T]$ denotes the expectation conditioned on the event $\{\tau \in T\}$ and $\mathbb{P}_\pi(\tau \in T) = \mathbb{E}_{\tau \sim \pi} [\Phi_k(\tau)]$.

Next, we would like to show that

$$\max_{\pi \notin \mathcal{S}} \mathbb{E}_{\tau \sim \pi} \left[ \sum_t r_t \right] - \mathbb{P}_\pi(\tau \notin T) \mathbb{E}_{\tau \sim \pi} \left[ \sum_t r_t \,\middle|\, \tau \notin T \right] < \max_\pi \mathbb{E}_{\tau \sim \pi} \left[ \sum_t r_t \right]. \tag{30}$$

Note that it is trivial that $\max_{\pi \notin \mathcal{S}} \mathbb{E}_{\tau \sim \pi} \left[ \sum_t r_t \right] - \mathbb{P}_\pi(\tau \notin T) \mathbb{E}_{\tau \sim \pi} \left[ \sum_t r_t \mid \tau \notin T \right] \leq \max_\pi \mathbb{E}_{\tau \sim \pi} \left[ \sum_t r_t \right]$ by Assumption 1 (i.e., $r_t \geq 0$). We just need to verify that the equality condition is infeasible. We prove by contradiction. In the case of equality, the left-hand side yields a global optimum $\pi^* \notin \mathcal{S}$. However, by Assumption 3 (i.e., $\forall \tau \sim \pi^*, \sum_{r_t \sim \tau} r_t > 0$), we have $\mathbb{E}_{\tau \sim \pi^*} [\sum_t r_t | \tau \notin T] > 0$. Since $\mathbb{P}_{\pi^*}(\tau \notin T) > 0$, a contradiction yields. Thus, Eq. (30) holds.

Accordingly, let

$$\Delta = \max_\pi \mathbb{E}_{\tau \sim \pi} \left[ \sum_t r_t \right] - \max_{\pi \notin \mathcal{S}} \left\{ \mathbb{E}_{\tau \sim \pi} \left[ \sum_t r_t \right] - \mathbb{P}_\pi(\tau \notin T) \mathbb{E}_{\tau \sim \pi} \left[ \sum_t r_t \middle| \tau \notin T \right] \right\} \quad (31)$$

and we can choose $\lambda \leq \frac{\Delta}{k\delta}$. With such a $\lambda$, we can continue the derivation in Eq. (29) by

$$\max_{\pi \notin \mathcal{S}} \text{RHS} < \max_{\pi \notin \mathcal{S}} \mathbb{E}_{\tau \sim \pi} \left[ \sum_t r_t \right] - \mathbb{P}_\pi(\tau \notin T) \mathbb{E}_{\tau \sim \pi} \left[ \sum_t r_t \middle| \tau \notin T \right] + \lambda k \delta$$
$$\leq \max_\pi \mathbb{E}_{\tau \sim \pi} \left[ \sum_t r_t \right] = \max_{\pi \in \mathcal{S}} \mathbb{E}_{\tau \sim \pi} \left[ \sum_t r_t \right]. \quad (32)$$

Therefore, we can conclude that the maximum value of the objective function in Eq. (24) can be only obtained from $\pi \in \mathcal{S}$. Finally, by induction, the solution of Eq. (24) must be a solution of Eq. (16).

□

**Corollary G.3.** *Providing Assumption 1, 2, and 3, the filtering objective and the switching objective can find $M$ distinct global optima in $M$ iterations.*

**Intuition Remark:** We want to emphasize that the constraint by $D_{\text{filter}}$ in Eq. (16) does not mean the infimum over *every possible* trajectories, but the infimum over a *subspace* of trajectories generated by the learning policy $\pi_{k+1}$. From the perspective of functional analysis, $\pi_{k+1}$ can produce 0 probability w.r.t. some state-action pairs, which restricts the space of possible trajectories sampled from $\pi_{k+1}$.

**Practical Remark 1:** Note that even though in practice, a neural network policy may never produce an action probability of zero due to approximation error, we would like to remark that the constraint-violated trajectories will have *a very low probability to be sampled*, that is, we typically have $\mathbb{P}_\pi(-\sum_t \log \pi(a_t|s_t) \geq \delta) \approx 1$ when the policy converges. In this case, the diversity constraints will be rarely violated given a *limited number* of trajectory *samples*. This is empirically justified as shown in Fig. 5 and Fig. 12 where the trajectory acceptance rate consistently stays at 1 in the later stage of training.

**Practical Remark 2:** Although the theorems only justify our algorithm when the optimal solutions are found, we empirically notice that our algorithm can even effectively discover a surprisingly diverse set of *local optimal* policies as shown in the experiment section. We believe there will be still huge room for further theoretical analysis, which we leave as future work.

# H ADDITIONAL DISCUSSIONS

## H.1 COMPARISON WITH CLASSICAL EXPLORATION METHODS

RSPO is beyond an exploration method. In the classical exploration literature, the goal is to discover a single policy that can approach the global optimal solution to produce the highest reward. By contrast, the goal of RSPO is not to just find a policy with high reward. Instead, RSPO aims to find as many distinct local optima as possible. In our experiments, a standard PPO trial in iteration 1 of RSPO can directly solve the task with the highest rewards. However, there are still a large number of distinct local optima with novel emergent behaviors other than this high-reward one. This is a particularly challenging constrained optimization problem as more local optima are discovered.

## H.2 SUBOPTIMALITY OF DISCOVERED POLICIES

It is possible that a policy produced by RSPO reaches a suboptimal solution if nearly-optimal solutions have been all discovered, such as in the MuJoCo domain. Meanwhile, RSPO could also possibly find

an optimal solution if the previously discovered solutions are all suboptimal, such as in the stag-hunt games. Overall, as a general solution, RSPO can definitely be applied as an exploration method by escaping sub-optimal strategies.

### H.3 MAY RSPO FAIL TO PRODUCE DIVERSE SOLUTIONS?

As more strategy modes are discovered, RSPO is solving an increasingly challenging constrained optimization problem, so it is indeed possible that an iteration "fails", e.g., some constraints may be violated or the policies may simply converge to previous modes leading. We empirically observe that these "failure" iterations typically lead to visually indistinguishable behaviors, which would not affect our final evaluation metric.

## I   LICENSES

We use MuJoCo under a personal license.

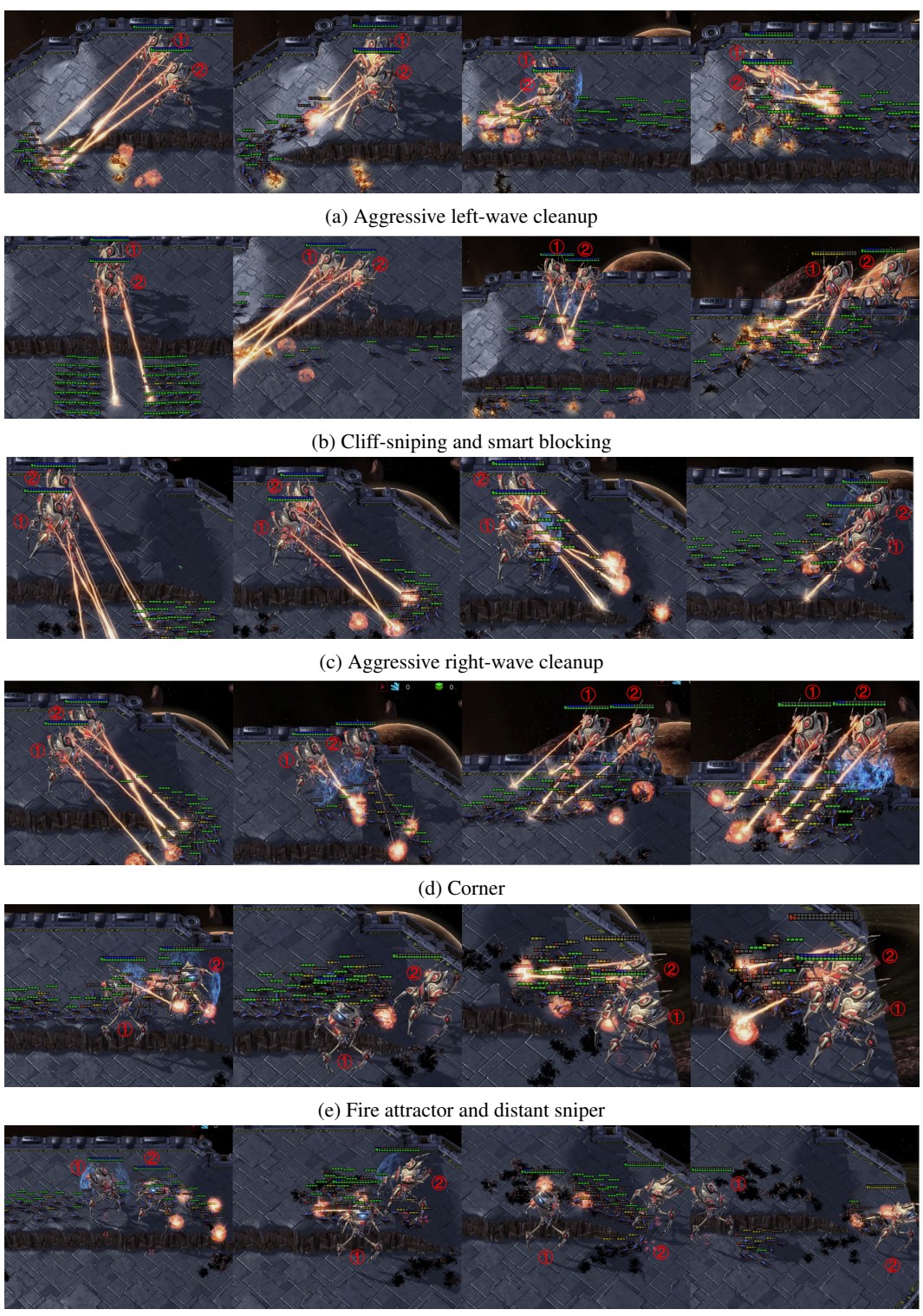

(a) Aggressive left-wave cleanup

(b) Cliff-sniping and smart blocking

(c) Aggressive right-wave cleanup

(d) Corner

(e) Fire attractor and distant sniper

(f) Cliff walk

Figure 14: Diverse strategies discovered by RSPO on the *2c_vs_64zg* map in SMAC.

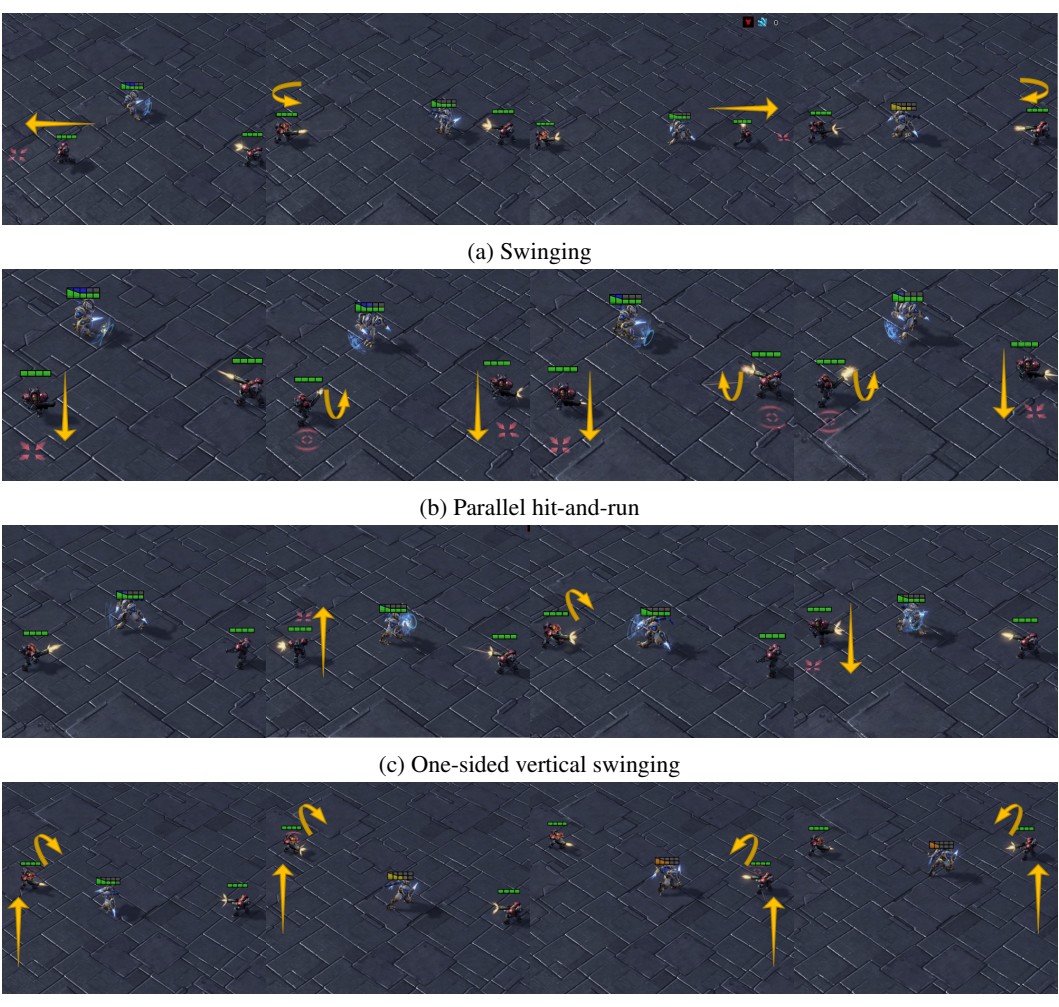

(a) Swinging

(b) Parallel hit-and-run

(c) One-sided vertical swinging

(d) Alternative distraction

Figure 15: Diverse strategies discovered by RSPO on the *2m_vs_1z* map in SMAC. The movement direction is indicated by yellow arrows.

