# OpenReview forum: "Continuously Discovering Novel Strategies via Reward-Switching Policy Optimization"
_ICLR.cc/2022/Conference — ICLR 2022 Poster_

### Official Review · Reviewer_kTab · 2021-10-27

**Correctness:** 4
**Technical Novelty And Significance:** 3
**Empirical Novelty And Significance:** 3
**Recommendation:** 8
**Confidence:** 3

**Main Review:**

Pro:
1. Simple approach that can generate policies with distinctive behaviors. Furthermore, a wide range of environments and many baseline methods are used to verify the effectiveness of the proposed approach.
2. The method is clearly described. The idea of trajectory filtering based on the sum of cross entropy and how to make use of rejected trajectories with the intrinsic reward to promote diversity is new to me.

Issue:
1. Some result indicates that the additional intrinsic reward can lead to suboptimal behaviors in terms of the extrinsic result. e.g., in the mujco environments, the 1st policy always demonstrates the best performance, and the subsequent policies can either perform worse or better than previous ones . It will be nice to provide a discussion about this.

**Summary Of The Paper:**

This paper presents a method to train diverse policies. New policies are trained to maximize the original reward while also optimized to be distinctive compared to previous policies. This is achieved by switching between optimizing for original reward and diversity reward. Multiple environments are used to demonstrate the proposed method is able to generate policies with distinctive behaviors.

**Summary Of The Review:**

This paper presents a simple and effective approach to generating policies with diverse behaviors. I believe this will be an important contribution to the field.

---

> ### Author Response · Authors · 2021-11-16
> **Author Response to Reviewer kTab**
>
> We would like to thank you for your appreciation and supportive comments.
>
> > the additional intrinsic reward can lead to suboptimal behaviors in terms of the extrinsic result... It will be nice to provide a discussion about this.
>
> + This is a great question. We would like to clarify that in the classical exploration literature, the goal is to discover a single policy that can approach the *global optimal solution* to produce the highest reward. By contrast, the goal of RSPO is not to just find a policy with high rewards. Instead, RSPO aims to find *as many distinct local optima as possible*. In our experiments, a standard PPO trial in iteration 1 of RSPO can directly solve the task with the highest rewards. However, there are still a large number of distinct local optima with novel emergent behaviors other than this high-reward one. This is a particularly challenging constrained optimization problem as more local optima are discovered.
>
> + Therefore, it is possible that a policy reaches a suboptimal solution if nearly-optimal solutions have been all discovered, such as in the MuJoCo domain. Meanwhile, RSPO could also possibly find an optimal solution if the previously discovered solutions are all suboptimal, such as in the stag-hunt games.
> + We have added the discussion in Appendix H.

---

### Official Review · Reviewer_tRht · 2021-10-28

**Correctness:** 2
**Technical Novelty And Significance:** 2
**Empirical Novelty And Significance:** 2
**Recommendation:** 5
**Confidence:** 3

**Main Review:**

### Strong points
- Proposed method finds the diverse strategies in the experiments

### Weak points
- It is not clear what is really optimized by the proposed algorithm
- design of the experiment is unclear and not convincing in some parts
- proposed algorithm seem to have many hyperparameters to be tuned

I describe the detailed comment to each point below.

### Meaning of the objective function
 I do not see a clear connection between the constrained optimization in Eq. (2) and (5). Although it is stated that the rejection-sampling-based approach enables to directly optimize Eq. (2) by strictly enforcing all the hard constraints, I don't understand why optimization problems in Eq. (2) and (5) are equivalent. It is necessary to show the equivalence mathematically.

In my understanding, the policy update is simply PPO, and the only difference from PPO is the use of the specific form of the reward function. In my opinion, the reward with the indicator function may encourage the policy not to overlap with other policies, but the policy update does NOT have any hard constraint.
In addition, the reward function in Eq. (9) also has the intrinsic reward which does not have the indicator function. The intrinsic reward seems to encourage the diverse behavior, but this term does not have any hard constraint. It is not clear what is really constrained in the  policy update using the proposed algorithm.

In addition, I'm not sure what is meant by "we also remark that trajectory filtering is closely related to the derivation of the
policy ratio clipping term in Proximal Policy Optimization (Schulman et al., 2017)." I encourage authors to mathematically  show the relation using equations.

If the authors cannot mathematically support the claims I pointed out above, these claims should be removed or modified.
In the current form, these claims are not properly supported, and the contribution is limited to just a reward engineering without theoretical justification.

### Experiment design
In the experiment section, the term "iteration" appears several times, but the definition of the iteration is not clear to me.
In my understanding, policies are trained sequentially in the proposed method, and I guess that a policy is optimized in each iteration. Then, I'm not sure how many environment steps are used in each iteration. If it requires significantly many steps, it is not fair to compare the results of keep running baseline methods. Rather, the baseline methods should be evaluated by repeating initializing and training the policy multiple times. For example, what happens if we run DIPG multiple times?  Another possible easy baseline is running SAC multiple times. (I guess that SAC is more sample efficient than simple policy gradient, and we can run SAC more times than PG with the same number of environment steps.)

### Hyperparameters
Judging from the appendix, hyperparameters such as $\lambda^{int}_B$, $\lambda^{int}_R$, $\alpha$, entropy-coefficient, PPO epochs, batch size are carefully tuned in the experiment. For fair comparison, hyperparameters of baseline methods should also be carefully tuned. In addition, the effect of some importance hyperparameter such as $\lambda^{int}_B$, $\lambda^{int}_R$, $\alpha$ should be analyzed.

Minor comments

- The proposed method is evaluated based on diversity metric, but the results of the average return based on the extrinsic reward should also be reported. As the goal of the proposed algorithm is to obtain the diverse policies that solves given tasks, the average return based on the extrinsic reward is also an important metric.

- I'm not sure whether the results in Tables 2 and 3 are the mean values over multiple trials or a single trial. If the results in Tables 2 and 3 are the mean values over multiple trials, the standard deviations should also be reported. Similar modifications should be made for Tables 6, 7, and 8 in Appendix as well.



**Summary Of The Paper:**

This paper presents an algorithm for discovering diverse strategies in reinforcement learning. In the proposed method, policies are sequentially trained with the constraint that encourages to find different behaviors. During the training process, the extrinsic and intrinsic rewards are switched based on a metric for quantifying the novelty of trajectories. The proposed method is evaluated on 4 domains, including OpenAI Gym Mujoco tasks and StarCraftII Multi-Agent Challenge. The experimental results show that the proposed method discovers diverse strategies in these domains.

**Summary Of The Review:**

Although the paper reports some interesting results, some of statements regarding the proposed algorithm do not seem well-supported.
In addition, the experimental section requires some improvement to justify the claims.

======== Comments after the author response ===========

I appreciate the authors' effort to address my concerns. During the conversation, the authors modified the several parts, which are crucial to support the problem formulation.  However, even after the in-depth discussion, I am not fully convinced that the claims are correctly proved.
Therefore, I keep the initial score.

---

> ### Author Response · Authors · 2021-11-16
> **Author Response to Reviewer tRht (1/2)**
>
> Thank you for your valuable comments! We have updated our paper to address the concerns.
>
> ## Meaning of the objective function
> > I do not see a clear connection between the constrained optimization in Eq. (2) and (5)... It is necessary to show the equivalence mathematically. … the reward function in Eq. (9) also has the intrinsic reward which does not have the indicator function. ...but this term does not have any hard constraint. It is not clear what is really constrained in the policy update using the proposed algorithm.
>
> + Please refer to Appendix G in our revised paper.
>
>
> > the reward with the indicator function may encourage the policy not to overlap with other policies, but the policy update does NOT have any hard constraint
>
> + Yes, there is no hard constraint during the learning process. However, with an effective RL algorithm such as PPO, the learned policy should gradually satisfy all the constraints as learning proceeds and eventually approaches a local optimum, as validated in Appendix G. Intuitively, the intrinsic rewards will be gradually turned off as more trajectories are accepted during training, which suggests the learning policies is satisfying more constraints. Please refer to Fig.5 and Fig.12 on the ratio of rejected samples, where we can observe that every trajectory of the RSPO policy is getting accepted at the end of training.
>
> + “Hard Constraints” in our paper means to enforce all the constraints after *approaching a local optimum*, instead of *at each gradient step*. Constraint violation at the beginning of the learning process may not affect the final performance of our algorithm.
>
>
> > I'm not sure what is meant by "we also remark that trajectory filtering is closely related to the derivation of the policy ratio clipping term in Proximal Policy Optimization (Schulman et al., 2017)."
>
> + We apologize for the confusion. There are two versions of PPO in the literature: one is to use a soft KL-divergence penalty in the learning objective[1] while the other is the clipped version[2]. In practice, the clipped version is substantially more robust and preferred by the community, which also motivates our algorithm, i.e., we also use a rejection sampling (analogous to "clipping") over trajectories rather than a soft objective. We have changed the sentence to “we also remark that trajectory filtering shares a conceptual similarity to the clipping term in Proximal Policy Optimization” in our revised version.
>
>
> ## Experiment Design
>
> > the definition of the iteration is not clear to me
>
> + RSPO adopts an iterative framework of repeatedly training individual policies, so “iteration” here refers to the process of training a new policy. Specifically, In iteration $k$ we train a new policy $\pi_k$ which is forced to be different from previously obtained $k - 1$ reference policies, i.e., $\pi_1$ to $\pi_{k-1}$. Hence, if we run RSPO for $K$ iterations, then we will obtain a set of $K$ distinct policies.
>
> > it is not fair to compare the results of keeping running baseline methods. Rather, the baseline methods should be evaluated by repeating initializing and training the policy multiple times
>
> + Baselines are indeed re-initialized and re-run multiple times with the same number of iterations as RSPO except for population-based methods. Population-based baselines are run for a single trial with a population of policies trained in parallel. We also make sure the population size is the same as the iteration number of RSPO. We have guaranteed all the algorithms are run for the same number of total environment steps and the same number of iterations (or population size). We have revised our paper to make this more clear.
>
> > Another possible easy baseline is running SAC multiple times. (I guess that SAC is more sample efficient than simple policy gradient)
>
> + We would like to remark that high sample efficiency is not sufficient to fully solve the multimodal exploration challenge in our testbeds. We have re-initialized and re-run PPO multiple times until convergence with different random seeds (please refer to PG results in our experiments). Without reward switching, PPO never discovered novel modes on the hard level of 4-goals and the stag-hunt games across all the repetitions. We argue that simple restarts would not be effective in highly complex multimodal problems.
>
> + We also want to remark that the focus of the paper is to discover diverse solutions using a minimal number of policies. To make the experiments fair, all the baselines and RSPO are built upon PPO and run for a long training process per iteration to ensure policies have converged.
>
> [1] Heess, N., TB, D., Sriram, S., Lemmon, J., Merel, J., Wayne, G., ... & Silver, D. (2017). Emergence of locomotion behaviours in rich environments. arXiv preprint arXiv:1707.02286.
>
> [2] Schulman, J., Wolski, F., Dhariwal, P., Radford, A., & Klimov, O. (2017). Proximal policy optimization algorithms. arXiv preprint arXiv:1707.06347.

---

> > ### Comment · Reviewer_tRht · 2021-11-16
> > **It is necessary to revise Section 3.3**
> >
> > > Yes, there is no hard constraint during the learning process.
> >
> > Then, it is necessary to modify the expressions in Section 3. For example, there is a sentence like
> > "we propose Trajectory Filtering, a rejection-sampling-based technique to directly optimize Eq. (2) by strictly enforcing all the hard constraints without adding extra terms to $J(\theta)$." I think this is significantly misleading.
> >
> > > However, with an effective RL algorithm such as PPO, the learned policy should gradually satisfy all the constraints as learning proceeds and eventually approaches a local optimum, as validated in Appendix G.
> >
> > > “Hard Constraints” in our paper means to enforce all the constraints after approaching a local optimum, instead of at each gradient step. Constraint violation at the beginning of the learning process may not affect the final performance of our algorithm.
> >
> > I'm not aware of such usages of the term "Hard Constraints" in the literature of RL. If we use the authors' definition of "Hard Constraints," the soft constraints in (4) can also be called the "Hard Constraints." That's too confusing, and the authors should avoid use the term "hard constraints."
> >
> > In addition, there is no guarantee that the policy reaches the optimum in the proposed algorithm.
> > We can prove the monotonic improvement of the policy for PPO in the same manner as TRPO. However, it does not mean that the policy converges to the optimal policy. The constraints will be satisfied at the optimum, but there is no guarantee that the policy reaches the optimum in practice. In other words, there is no guarantee that the constraints are satisfied even if PPO is used to optimize the policy in the proposed algorithm.
> >
> > >Please refer to Appendix G in our revised paper.
> >
> > I appreciate the authors' effort for showing the connection between (2) and the reward function in (9). However, if I understand correctly, the discussion in Appendix G revealed that the following things:
> > - the proposed algorithm is actually based on the soft constraint, although the current logic in Section 3.3 sounds like the proposed algorithm is based on hard constraints
> > - the solution of the problem formulated in (2) can be approximated by maximizing the reward given by (9), and the problem in (2) is not equivalent to the problem in (5).
> >
> > If my understanding is correct, I think that the Section 3.3 and 3.4 should be revised significantly.
> > In the current form, it sounds like the problem in (2) is equivalent to the problem in (5), and the objective in (6) is a kind of auxiliary objective to leverage the rejected trajectories. However, it seems that the combination of the objectives in (5) and (6) are equivalent to the problem in (2).
> > Maybe authors can consider replacing the large part of Section 3.3 with the discussion in Appendix G.
> >
> > I'm happy to raise the score if my concerns are appropriately addressed. Please let me know whether my understanding is correct or not and how my concerns will be resolved.

---

> > > ### Author Response · Authors · 2021-11-17
> > > **thank you for your prompt response and constructive suggestion**
> > >
> > > We thank you for your prompt response and constructive suggestion.
> > >
> > > First, we now fully get your point and agree that our original statements using the term “hard constraint” can be confusing. We would like to clarify that the purpose of using the term “hard” is simply to contrast the term “soft objective”, i.e., the Lagrangian multiplier approach. In fact, our method is a *filtering-based* approach. We have revised our paper and replaced the term “hard” with “filtering-based” in Sec 3.3.
> > >
> > > Furthermore, we would like to provide some further clarifications on the equations. Both Eq. (5) and Eq. (6) are equivalent to Eq. (2) according to Lemma G.1 and Lemma G.2 respectively. Eq. (9) is also equivalent to Eq. (2) when we use the behavior intrinsic reward $r^{int}_B$ solely. For better practical performance, we also additionally introduce another reward-prediction-error-based intrinsic reward $r^{int}_R$ as an empirical suggestion. We remark that in most of our experiments, only using the behavior reward is sufficient. We have provided more discussions on the equations in Sec 3.4. Please check our revised paper for details.

---

> > > > ### Comment · Reviewer_tRht · 2021-11-19
> > > > **one more comments**
> > > >
> > > > Thank you for the clarification and the update of the paper. I would like to ask a few more points.
> > > >
> > > > Still I'm not sure why Lemma 5.1 shows that the solution of Eq. (5) is equivalent to that of Eq. (2). The form of (19) is still different from (5). In (5), we have the product of the indicator function and the reward, while we have the separate terms of the expected reward and the divergence metric in (19). Please elaborate the connection.
> > > >
> > > > In Apppendix G, there is a statement "By further estimating the expectation using trajectory samples, we can derive
> > > > the trajectory filtering objective in Eq. (5) using Lemma G.1 and the reward switching objective in
> > > > Eq. (6) using Lemma G.2." I think this is the important part. Please write down the derivation with equations.

---

> > > > > ### Author Response · Authors · 2021-11-21
> > > > > **Please check Appendix G in our revised paper**
> > > > >
> > > > > We have rewritten Appendix G for better readability and completeness. Please let us know if you have any further questions.

---

> > > > > > ### Comment · Reviewer_tRht · 2021-11-22
> > > > > > **Another question**
> > > > > >
> > > > > > Regarding the constraint, I don't understand the meaning of $\textrm{inf}_{(s,a) \sim \tau} \sum_t \log \pi_i (a|s) > \delta$, which I think was in a different form in the previous version.
> > > > > > Why do we need to consider the infimum of $\sum_t \log \pi_i (a|s)$ ?
> > > > > >
> > > > > > Regarding Appendix F., I also don't understand why we need to compute the infimum of the KL divergence. I think the KL divergence is often approximated using the expectation as $\int p(s) \log \frac{p(x)}{q(x)} dx = \mathbb{E}[ \log p(x) - \log q(x) ]$, but I'm not sure how the infimum comes into play.
> > > > > > Can you provide any references regarding this approximation?

---

> > > > > > > ### Author Response · Authors · 2021-11-22
> > > > > > > **Author response for clarification**
> > > > > > >
> > > > > > > We apologize for all the presentation confusion. The infimum comes from our proof in Appendix G. Our theorem shows that solving Eq.(5) is corresponding to solving Eq.(2) w.r.t. an even stronger diversity measure, i.e., the infimum of trajectory negative likelihood. For the best readability and presentation consistency, we have moved all the related content to the appendix and updated appendix F & G accordingly. We hope the current version is much easier to follow.
> > > > > > >
> > > > > > > Intuitively, using the infimum constraint implies *each trajectory* generated by policy $\pi_k$ should be sufficiently novel (rather than in expectation). In practice, RSPO ignores the extrinsic rewards on *any* trajectories violating the novelty constraint. The diversity intrinsic reward on rejected trajectories will constantly push the learning policy towards the feasible policy space and the reward-switching objective will eventually converge to the extrinsic return when no trajectory is rejected --- this implies the learning policy has approximately satisfied the infimum constraint. We also remark that some existing work[1] presents a similar formulation by constraining the safety signal *at each state*, which could be seen as an infimum constraint over all states.
> > > > > > >
> > > > > > > [1] Dalal, G., Dvijotham, K., Vecerik, M., Hester, T., Paduraru, C., & Tassa, Y. (2018). Safe exploration in continuous action spaces. arXiv preprint arXiv:1801.08757.

---

> > > > > > > > ### Comment · Reviewer_tRht · 2021-11-24
> > > > > > > > **One more (hopefully, last) comment**
> > > > > > > >
> > > > > > > > I  think that the infimum in (16) should be the expectation. There is a gap between problems in (16) and (19).
> > > > > > > > The unconstrained optimization problem in (19) is defined as maximization of the expectation with respect to the trajectories drawn by the policy. Meanwhile, the constraint in (16) indicates that any trajectory generated by the policy needs to satisfy the constraint.
> > > > > > > > I don't think that we can obtain such a guarantee as long as a policy is stochastic and we solve the problem based on the expectation as in (19).

---

> > > > > > > > > ### Author Response · Authors · 2021-11-24
> > > > > > > > > **Further clarificaiton**
> > > > > > > > >
> > > > > > > > > First, we would like to emphasize that in Eq. (19), there is an indicator function $I_i (\tau)$ associated with *each* trajectory $\tau$, which is in a very different form from the standard Lagrangian multiplier where a constant coefficient is applied to all the trajectories. This indicator function ensures that the converged solution satisfies the infimum constraint rather than the expectation. Please check our proof on the top of page 25.
> > > > > > > > >
> > > > > > > > > Furthermore, we want to point out that, *theoretically*, even if the optimal policy is stochastic, the infimum constraint can be still satisfied. The key intuition is that the optimal policy may not generate *every* possible trajectory --- some trajectories can just have a zero likelihood. Let’s consider a motivating example to illustrate this intuition, where an agent spawns at the center and navigates in a $N\times N$ grid world. Assume policy $\pi_1$ only has a nonzero probability of moving forward and moving left, and policy $\pi_2$ only has a nonzero probability of moving backward and moving right. Then both $\pi_1$ and $\pi_2$ are stochastic but the infimum constraint is $+\infty$. In general, the constraint can be satisfied if we can arbitrarily choose a policy $\pi$, such that $\pi$ only covers an (almost) disjoint trajectory subspace.
> > > > > > > > >
> > > > > > > > > We want to remark that even though in practice, a neural network policy may never produce an action probability of zero due to approximation error,  the constraint-violated trajectories will have *a very low probability to be sampled*, i.e., $\mathbb{P}(-\sum_t \log\pi (a_t|s_t)\ge\delta)\approx 1$. This is also justified by the evidence that the acceptance rate consistently stays at $1$ in the late stage of training, as shown in Fig.5 and Fig.12.
> > > > > > > > >
> > > > > > > > > We hope our clarifications could help. We sincerely appreciate the discussion and your feedback, which consistently makes our paper more clear.

---

> > > > > > > > > > ### Comment · Reviewer_tRht · 2021-12-01
> > > > > > > > > > **proof in Appendix G has a weird point**
> > > > > > > > > >
> > > > > > > > > > I'm not sure the proof in Appendix G (top of page 25) is correct or not. Something is wrong with the transition from line 1 to 2.
> > > > > > > > > >
> > > > > > > > > > I assume that $E_{\tau \sim \pi}[X] = \int p(\tau; \theta) X d\tau,  E_{\tau \sim \pi}[X|\tau \in T] = \int p_{\in T}(\tau; \theta) X d\tau, E_{\tau \sim \pi}[X|\tau \notin T] = \int p_{\notin T}(\tau; \theta) X d\tau$.
> > > > > > > > > > Here, $p_{\in T}(\tau; \theta)$ represents the probability density of the trajectories that satisfy the constraint, and $p_{\notin T}(\tau; \theta)$ represents the probability density of the trajectories that do not satisfy the constraint.
> > > > > > > > > >
> > > > > > > > > > Then, $E_{\tau \sim \pi}[X] = \int p(\tau; \theta) X d\tau = \int ( p_{\in T}(\tau; \theta) + p_{\notin T}(\tau; \theta))  (\tau; \theta) X d\tau = E_{\tau \sim \pi}[X|\tau \in T] +   E_{\tau \sim \pi}[X|\tau \notin T] $.
> > > > > > > > > >
> > > > > > > > > > However, $\mathbb{P}(\tau \in T)$ and $\mathbb{P}(\tau \notin T)$ additionally appear in line 2.
> > > > > > > > > > Based on a educated guess, I assume that $\mathbb{P}(\tau \in T)$ is the probability that the trajectory generated by $\pi$ satisfies the constraint, which is given by $\mathbb{P}(\tau \in T)=\int p_{\in T}(\tau; \theta)  d \tau$, and that $\mathbb{P}(\tau \in T) + \mathbb{P}(\tau \notin T) = 1$.
> > > > > > > > > >
> > > > > > > > > > To me, it does not make sense to have these additional terms in line 2. It is essential to clarify the definition of $\mathbb{P}(\tau \in T)$ and related terms and/or modify the equations if necessary.

---

> > > > > > > > > > > ### Author Response · Authors · 2021-12-01
> > > > > > > > > > > **Thanks for your responsive and consistent feedback across the rebuttal period**
> > > > > > > > > > >
> > > > > > > > > > > We would like to remark that for any event $A$, the following equation generally holds:
> > > > > > > > > > > $E[X]=P(A)E[X|A]+P(A^c)E[X|A^c]$, where $A^c$ denotes the [*complement*](https://en.wikipedia.org/wiki/Complement_(set_theory)) of $A$.
> > > > > > > > > > > This equation is called *the law of total expectation*. Please refer to the [wikipedia page](https://en.wikipedia.org/wiki/Law_of_total_expectation) for rigorous proof.
> > > > > > > > > > >
> > > > > > > > > > > Let’s consider a concrete example here: given a random variable $X$ following the uniform distribution in $[0, 1]$, let $A$ be $0\le X<0.9$ and $A^c$ be $0.9\le X\le 1$. Then we could compute the conditional expectation by $E[X|A]=0.45$ and $E[X|A^c]=0.95$. Therefore,
> > > > > > > > > > > $E[X]=0.5=0.9E[X|A]+0.1E[X|A^c]\ne E[X|A]+E[X|A^c]$.
> > > > > > > > > > >
> > > > > > > > > > > We are more than happy to make further clarifications if you have any concerns.

---

> > > > > > > > > > > > ### Comment · Reviewer_tRht · 2021-12-02
> > > > > > > > > > > > **Let me elaborate the question**
> > > > > > > > > > > >
> > > > > > > > > > > > In my previous question, the definition of $E_{\tau \sim \pi}[X|\tau \in T]$ I used is not the conventional conditional expectation. It is tricky because we are computing the integral over $\tau$, which is also conditioned on $\tau$.
> > > > > > > > > > > > Of course, $E[X] \neq E[X|A] + E[X|A^c]$ in general. Actually, the definition of $E_{\tau \sim \pi}[X|\tau \in T]$ is the crucial point.
> > > > > > > > > > > >
> > > > > > > > > > > > I am asking the question because the definition of $E_{\tau \sim \pi}[X|\tau \in T]$ is not clear.
> > > > > > > > > > > > As the trajectory $\tau$ is continuous, $E_{\tau \sim \pi}[X|\tau \in T]$ should be given by the integral over $\tau$. However, the definition of $E_{\tau \sim \pi}[X|\tau \in T]$ is unclear, and it is not clear how the equations are obtained in the proof.
> > > > > > > > > > > >
> > > > > > > > > > > > As $\tau$ is continuous, it is more natural to consider $\mathbb{E}[X] = \int p(y) \mathbb{E}[X|y] dy$ rather than $\mathbb{E}[X] = \sum \mathbb{P}(A) \mathbb{E}[X|A]$. Of course, we can easily connect them using something like $\int_{A} p(x) dx = \mathbb{P}(A)$,  but it is necessary to show the definition of $E_{\tau \sim \pi}[X|\tau \in T]$ by defining the necessary probability density functions.

---

> > > > > > > > > > > > > ### Author Response · Authors · 2021-12-02
> > > > > > > > > > > > > **X is a function over tau in our proof**
> > > > > > > > > > > > >
> > > > > > > > > > > > > Now we fully get your point.
> > > > > > > > > > > > >
> > > > > > > > > > > > > We would like to clarify that in line 1 of our proof at page 25, $X$ is a function over $\tau$ directly following the right hand side of Eq. (19) just for conciseness. In particular, $X$ is defined by
> > > > > > > > > > > > > $$X=X(\tau)=\prod_{i=1}^{k} I_i (\tau)\sum_t r_t +\sum_{i=1}^{k}\left(1-I_i (\tau)\right)\left(-\sum_t \log \pi_i(a_t|s_t)\right).$$
> > > > > > > > > > > > > Thus, given an event $A: \{ \tau\in T \}$, $E_{\tau\in\pi}[X]=E_{\tau\in\pi}[X(\tau)]=P(A)E_{\tau\in\pi}[X(\tau)|A]+P(A^c)E_{\tau\in\pi}[X(\tau)|A^c]$, which follows the standard conditional expectation derivation. We remark that the condition is *NOT* over the random variable $\tau$. Instead, it is [conditioned over the **event**](https://en.wikipedia.org/wiki/Conditional_expectation#Conditioning_on_an_event) $A: \{\tau\in T\}$.
> > > > > > > > > > > > >
> > > > > > > > > > > > > The reason we use $E_{\tau}[X(\tau)]=P(A)E_{\tau}[X(\tau)|A]+P(A^c)E_{\tau}[X(\tau)|A^c]$ is because $\{A,A^c\}$ is a finite partition of the sample space over $\tau$, which follows the conventional notation by [the law of expectation](https://en.wikipedia.org/wiki/Law_of_total_expectation).
> > > > > > > > > > > > >
> > > > > > > > > > > > > We will replace $X$ in line 1 and line 2 from our proof with a functional notation like $g(\tau)$ in our final revision.
> > > > > > > > > > > > >
> > > > > > > > > > > > > We apologize for the possible confusion and hope this could address your concerns.

---

> > > > > > > > > > > > > ### Comment · Area_Chair_nSRT · 2021-12-02
> > > > > > > > > > > > > **Another pair of eyes**
> > > > > > > > > > > > >
> > > > > > > > > > > > > Dear Reviewer tRht and authors,
> > > > > > > > > > > > >
> > > > > > > > > > > > > I really appreciate this in-depth exchange as I believe it is indeed important to validate the correctness of the theoretical analysis of the method.
> > > > > > > > > > > > >
> > > > > > > > > > > > > I agree with Reviewer tRht that the conditional expectation $E_{\tau \sim \pi}[X | \tau \in T]$ should be clearly defined, as otherwise it can be a source of confusion (as illustrated by this long discussion). My own understanding -- authors please correct me if I'm wrong -- is that
> > > > > > > > > > > > > $$E_{\tau \sim \pi}[X | \tau \in T] = \int_{\tau \in T} X \pi(\tau | \tau \in T) d\tau$$
> > > > > > > > > > > > >
> > > > > > > > > > > > > I would also suggest to write $P_{\theta}(\tau \in T)$ instead of $P(\tau \in T)$ and explicitly define it as being equal to $E_{\tau \sim \pi_{\theta}}[1_{\tau \in T}]$. Otherwise it is confusing (initially I didn't understand what it meant in eq. 18), and it is dangerously easy to forget that it depends on $\theta$.
> > > > > > > > > > > > >
> > > > > > > > > > > > > Another small point regarding the proof is on eq. 23, which differs from eq. 19 not just by the absence of intrinsic rewards, but the fact that a conditional expectation is used while eq. 19 uses a non-conditional expectation, which isn't equivalent even with the additional indicator functions (unless my understanding above is incorrect). As a result, the proof of Corollary G.2 that just says "follow the proof of Theorem G.1" isn't entirely obvious. I wonder if maybe eq. 23 was meant to be written differently, especially since the text below mentions the $I_i(\tau)$ even though they seem irrelevant to Corollary G.2 in its current formulation.
> > > > > > > > > > > > >
> > > > > > > > > > > > > I do have one major concern with Theorem G.1 though, which is that it relies on an unverifiable assumption that seems to make the conclusion trivially true (eq. 18). Indeed, it seems to me that it basically amounts to saying "if the policies that don't satisfy the constraint are bad enough, then the best policy of the unconstrained problem is satisfying the constraints". This makes this theoretical result much less interesting IMO (why would the policies not satisfying the constraint be bad in the first place?)
> > > > > > > > > > > > >
> > > > > > > > > > > > > Regarding Corollary G.2, I will wait for the authors' confirmation of eq. 23 before looking into it more closely, but I'm a bit puzzled by eq. 22 which appears to me to be always true if $r_t > 0$ on each timestep. I'm not entirely sure what to make of it, since the sign of $r_t$ isn't something that seems to be discussed anywhere. I will give it more thought and elaborate on it later, but if authors have some insights on this point, I'm curious to hear them.
> > > > > > > > > > > > >
> > > > > > > > > > > > > One last comment on the theory is that there are some shortcuts being taken, with the discount factor $\gamma$ disappearing, and no indication of the length of trajectories (do they have fixed length? can they be infinite?). I believe it makes sense to make some simplifications in order to keep the theory digestible, but it would be better if these simplifications were stated explicitly.
> > > > > > > > > > > > >
> > > > > > > > > > > > > I would appreciate if the authors could comment on these points -- especially if there is something I misunderstood.
> > > > > > > > > > > > >
> > > > > > > > > > > > > Thanks!
> > > > > > > > > > > > >
> > > > > > > > > > > > > -- Your AC
> > > > > > > > > > > > >
> > > > > > > > > > > > > PS: actually, a few mother minor points:
> > > > > > > > > > > > > - In the definition of D_filter at beginning of Section G, the order of pi_i and pi_j seems reversed compared to the order used in eq. 3 => it would be better to use a consistent order
> > > > > > > > > > > > > - Could phi_i be used instead of I_i to keep notations consistent between the main text and the appendix? (unless there's a subtle difference I'm missing)
> > > > > > > > > > > > > - In the proof of Th. G1, the sum with the log \pi_i should be a sum over t and not over i
> > > > > > > > > > > > > - Although my main concern is the one related to the assumption in eq. 18, I should mention that I am also a bit unconvinced by the argument at the bottom of p. 28 trying to justify the infimum, considering that in practice we typically train policies that have non-zero probability everywhere. It is unclear why "near-zero" would be enough, without some proper mathematical justification, or at the very least an intuitive explanation.

---

> > > > > > > > > > > > > > ### Author Response · Authors · 2021-12-03
> > > > > > > > > > > > > > **Author Response to AC**
> > > > > > > > > > > > > >
> > > > > > > > > > > > > > We sincerely thank your comments regarding our theoretical analysis in Appendix G.
> > > > > > > > > > > > > >
> > > > > > > > > > > > > > We have carefully re-examined the entire Appendix G and substantially rewritten the section. We deeply apologize for all the confusion in the early version of Appendix G.
> > > > > > > > > > > > > >
> > > > > > > > > > > > > > The URL to our updated draft is https://drive.google.com/file/d/1MAQSsKZuE85k-j7K8lWRprE2LyAsA_8D/view?usp=sharing, which is also posted at our anonymous [project website](https://sites.google.com/view/rspo-iclr-2022) (see Appendix A).
> > > > > > > > > > > > > >
> > > > > > > > > > > > > > We emphasize that none of the theoretical changes affect our actual algorithm. All the implementation details, empirical results, intuitions, and conclusions remain unaffected.
> > > > > > > > > > > > > >
> > > > > > > > > > > > > > Some of the highlighted changes and clarifications are summarized below.
> > > > > > > > > > > > > >
> > > > > > > > > > > > > > ### Changes
> > > > > > > > > > > > > > 1. We have changed Eq.(5) from a conditional expectation back to the original unconditioned expectation with the indicator function.
> > > > > > > > > > > > > > 2. We add the missing $\lambda$ (the coefficient of intrinsic rewards) back to Eq.(6)
> > > > > > > > > > > > > > 3. Appendix G is substantially rewritten with general and intuitive assumptions and rigorous proofs.
> > > > > > > > > > > > > >
> > > > > > > > > > > > > > ### Clarifications:
> > > > > > > > > > > > > > > regarding the definition of $E_{\tau\sim\pi}[X|\tau\in T]$
> > > > > > > > > > > > > >
> > > > > > > > > > > > > > Thanks for pointing this out. We have changed all the notations accordingly in the updated draft.
> > > > > > > > > > > > > >
> > > > > > > > > > > > > >
> > > > > > > > > > > > > > > As a result, the proof of Corollary G.2 that just says "follow the proof of Theorem G.1" isn't entirely obvious.
> > > > > > > > > > > > > >
> > > > > > > > > > > > > > The corresponding objective has been changed back to the form of the indicator function.
> > > > > > > > > > > > > >
> > > > > > > > > > > > > >
> > > > > > > > > > > > > > > eq. (18)...seems to me that it basically amounts to saying "if the policies that don't satisfy the constraint are bad enough, then the best policy of the unconstrained problem is satisfying the constraints"... why would the policies not satisfy the constraint be bad in the first place?
> > > > > > > > > > > > > >
> > > > > > > > > > > > > > We have substantially changed the assumptions in our paper. New assumptions are more general and intuitive.
> > > > > > > > > > > > > >
> > > > > > > > > > > > > >
> > > > > > > > > > > > > > > One last comment on the theory is that there are some shortcuts being taken…
> > > > > > > > > > > > > >
> > > > > > > > > > > > > > Yes, we do assume a fixed horizon. All the simplifications have been stated explicitly in the new Appendix G.
> > > > > > > > > > > > > >
> > > > > > > > > > > > > >
> > > > > > > > > > > > > > > the order of pi_i and pi_j seems reversed…
> > > > > > > > > > > > > >
> > > > > > > > > > > > > > > Could phi_i be used instead of I_i to keep notations consistent between the main text and the appendix…
> > > > > > > > > > > > > >
> > > > > > > > > > > > > > > In the proof of Th. G1, the sum with the log \pi_i should be a sum over t and not over i…
> > > > > > > > > > > > > >
> > > > > > > > > > > > > > All the typos are fixed.
> > > > > > > > > > > > > >
> > > > > > > > > > > > > > > I should mention that I am also a bit unconvinced by the argument at the bottom of p. 28 trying to justify the infimum…
> > > > > > > > > > > > > >
> > > > > > > > > > > > > > We have put a few practical remarks on the gap between theory and practice at the end of the new appendix G.
> > > > > > > > > > > > > >
> > > > > > > > > > > > > > It is true that in practice, a neural network policy may never produce a policy with an action probability of exactly 0. However, many trajectories can be rarely generated through a limited number of samples. So, in fact, the infimum can be approximately satisfied as empirically justified in Fig. 5 and Fig. 12 in our paper, where we can observe that the trajectory acceptance rate consistently stays at $1$ in the later stage of training.
> > > > > > > > > > > > > >
> > > > > > > > > > > > > > ### Final Remark
> > > > > > > > > > > > > > We again apologize for the confusion in the theoretical part of our work, but we would like to kindly remark that the contribution of our paper is much beyond the theorem in Appendix G. RSPO is itself a novel and interesting paradigm that can discover diverse strategies across a wide range of challenging RL domains in practice. We hope our empirical results could benefit the community and possibly lead to more interesting theoretical insights.
> > > > > > > > > > > > > >
> > > > > > > > > > > > > > Best,
> > > > > > > > > > > > > >
> > > > > > > > > > > > > > Paper2821 Authors

---

> > > > > > > > > > > > > > > ### Comment · Area_Chair_nSRT · 2021-12-03
> > > > > > > > > > > > > > > **Re: Author Response to AC**
> > > > > > > > > > > > > > >
> > > > > > > > > > > > > > > Dear authors,
> > > > > > > > > > > > > > >
> > > > > > > > > > > > > > > Thank you for the extensive answer to my comments. I will just mention a few small points to conclude this discussion (I do not expect a reply on these given the tight timeline, but I thought it would be worth sharing with you):
> > > > > > > > > > > > > > > * I think that the relevance of the sign of r_t is a hint that there may be something to improve on. Intuitively, looking at eq. 6, it seems like there might be a risk of the optimal solution being a policy that is sitting "on the edge of the intended constraints", i.e., that would generate invalid trajectories with phi(tau) = 0 while maximizing the intrinsic rewards. It would be interesting to check whether this can actually be an issue in practice, and if yes, how to solve it in a general way.
> > > > > > > > > > > > > > > * Although shifting rewards by a constant offset (to make them >0) is always possible in theory, it can cause optimization issues in practice. It is unclear here to which extent this is something that should be considered when applying this technique, and how to achieve it in a way that works well.
> > > > > > > > > > > > > > > * Maybe use $\in$ rather than the equality sign to say that a policy is optimal since you are assuming multiple optima (e.g., $\pi_i^* \in \mathbb{argmax}_{\pi} ...$)
> > > > > > > > > > > > > > > * Beginning of proof of Th. G1, "the optimization problems (...) are trivial" => "are trivially equivalent" (since solving them isn't trivial)
> > > > > > > > > > > > > > > * Eq. 22: I think the RHS should be a max over $\pi \in S$.
> > > > > > > > > > > > > > > * $P_{\pi^*}(\tau \notin T) > 0$ may not necessarily be guaranteed, since with continuous states / actions you can have zero probability mass even if there exist a few violating trajectories. Some additional assumptions / explanations might be required.
> > > > > > > > > > > > > > > * On the line above eq. 23 there is a dot in front of a sum that can be removed
> > > > > > > > > > > > > > > * The dot at end of eq. 24 can also be removed
> > > > > > > > > > > > > > >
> > > > > > > > > > > > > > > Best,
> > > > > > > > > > > > > > >
> > > > > > > > > > > > > > > Your AC

---

> > > > > > > > > > > > > > > > ### Author Response · Authors · 2021-12-04
> > > > > > > > > > > > > > > > **Re: Re: Author Response to AC**
> > > > > > > > > > > > > > > >
> > > > > > > > > > > > > > > > Dear AC,
> > > > > > > > > > > > > > > >
> > > > > > > > > > > > > > > > We apologize for the typos. All the typos have been corrected and we have also added an additional assumption that assumes finite states and actions.
> > > > > > > > > > > > > > > >
> > > > > > > > > > > > > > > > Here are some further clarifications/explanations:
> > > > > > > > > > > > > > > >
> > > > > > > > > > > > > > > > > shifting rewards by a constant offset (to make them >0) can cause optimization issues in practice.
> > > > > > > > > > > > > > > >
> > > > > > > > > > > > > > > > >  I think that the relevance of the sign of r_t is a hint that there may be something to improve on
> > > > > > > > > > > > > > > >
> > > > > > > > > > > > > > > >
> > > > > > > > > > > > > > > > Although the non-negative-reward assumption is required in the current form of our theoretical analysis, it does not affect the practical use of RSPO.
> > > > > > > > > > > > > > > >
> > > > > > > > > > > > > > > > On one hand, we use actor-critic methods (PPO) in practice, so the baseline (and advantage normalization) will naturally cancel the offset. The non-negative reward assumption is only for the purpose of proof and our method works well for domains with negative rewards in the experiment section.
> > > > > > > > > > > > > > > >
> > > > > > > > > > > > > > > > In addition, we fully agree that the theoretical results can be much improved and the analysis should not depend on any reward assumptions. Intuitively, RSPO only computes gradients over novel trajectories, which will enforce the policy to keep moving towards a novel policy that only generates constraint-satisfying trajectories (i.e., the infimum metric). This is the underlying intuition that motivates us to develop RSPO. Ideally, the most informative theory would be a convergence analysis on the entire gradient descent process, which is challenging but we do find it interesting and are working on it towards future work.
> > > > > > > > > > > > > > > >
> > > > > > > > > > > > > > > >
> > > > > > > > > > > > > > > > > Intuitively, looking at eq. 6, it seems like there might be a risk of the optimal solution being a policy that is sitting "on the edge of the intended constraints",
> > > > > > > > > > > > > > > >
> > > > > > > > > > > > > > > > In fact, “a solution on the edge” is a flaw for eq. (2). In the conventional formulation of constrained optimization and solutions using a Lagrangian multiplier, it is very possible that the converged solution is on the edge of constraints, which violates our mission of finding local optima. This motivates us to develop a new framework to enforce the local optimality.
> > > > > > > > > > > > > > > >
> > > > > > > > > > > > > > > > In Eq.(6), an interesting fact is that the intrinsic reward and the distance metric are in the same form, so as the intrinsic reward decreases to $\delta$, the learning policy naturally satisfies the constraints (if $\lambda$ is properly set). Then, RSPO reverts back to normal PG in the original MDP to ensure the converged policy is indeed a local optimum.
> > > > > > > > > > > > > > > > Intuitively, RSPO uses an intrinsic reward to push the policy moving towards a constrained-satisfying subspace and when the policy converges, it should be fully located within a feasible neighborhood of a novel local optimum. This is empirically justified by the trajectory acceptance rate illustrated in Fig. 5 and Fig. 12 --- when RSPO converges, the algorithm is indeed performing standard PG since all the samples are accepted.
> > > > > > > > > > > > > > > >
> > > > > > > > > > > > > > > > On a high level, we would like to clarify that our mission remains to solve Eq. (1). However, we notice that the conventional iterative paradigm (eq.(2)) with the cross-entropy metric may not ensure the policy converges to distinct local optima. Thus, we developed RSPO, which strictly enforces the constraints via rejection sampling. Our theoretical analysis suggests that RSPO is essentially tackling a problem with an even hard constraint (i.e., infimum), which explains why RSPO can discover visually diverse strategies while other baselines cannot. It is possible that RSPO may fail to fully satisfy the infimum constraints (e.g. pi(tau)=0 but tau violates the constraint) but empirically, the method still works well and the visualized strategies remain sufficiently diverse.
> > > > > > > > > > > > > > > >
> > > > > > > > > > > > > > > > Yet the current theoretical justifications have not fully revealed the potential of RSPO, we are consistently working on this and hope the strong empirical performance could bring up insightful discussions in the community.
> > > > > > > > > > > > > > > >
> > > > > > > > > > > > > > > > Best,
> > > > > > > > > > > > > > > >
> > > > > > > > > > > > > > > > Paper2821 Authors

---

> ### Author Response · Authors · 2021-11-16
> **Author Response to Reviewer tRht (2/2)**
>
> ## Hyperparameters
> Regarding hyperparameters, we would like to emphasize the following 3 points.
>
> 1. All the PPO-related hyperparameters are fixed across all the baselines.
>
> Now that all the baselines are built upon PPO, we fix these PPO-related hyperparameters across all the experiments for a fair comparison. We have revised our paper to make it more clear.
>
> 2. We have performed an extensive grid search over algorithm-specific hyperparameters.
>
> For PBT-CE, we performed a grid search over 1e0, 5e-1, 1e-2, 5e-2, 5e-3, 2e-3, and 1e-3 on the Lagrange multiplier in each domain, and only 1e-3 converges. For DIPG, we performed a grid search over 2e0, 1e0, 5e-1, 1e-1, and 1e-2 on the coefficient of MMD loss in each domain. For TrajDiv, we performed a grid search over 1e-2, 1e-1, 1e0 on the coefficient on the TrajDiv loss and over 1e-1, 5e-1, 9e-1 on the action-kernel discounting factor. For other baselines, the algorithm-specific hyperparameters are inherited from other papers which proposed the algorithm or utilized the algorithm as a baseline. We have clarified this in Appendix D.2.
>
> 3. We additionally performed a sensitivity analysis over $\lambda_R$, $\lambda_B$, and $\delta$ (or $\alpha$ for automatic threshold selection). Please refer to Appendix B.3 in our revised version.
>
> ## Others
> > the results of the average return based on the extrinsic reward should also be reported
>
> + For 4-goals, please refer to Fig 1(b). The extrinsic return of RSPO at the easy and medium level is 1.
> For Monster-Hunt, please refer to Fig 4. For Escalation, please refer to Fig 7(a).
> For MuJoCo and SMAC, please refer to Table 6 and Table 7 in Appendix B.2. We report the winning rate in SMAC since it is the most commonly accepted metric for evaluating performance in this environment.
>
> > I'm not sure whether the results in Tables 2 and 3 are the mean values over multiple trials or a single trial.
>
> + Results are over the set of policies obtained within a single experiment trial.

---

> > ### Comment · Reviewer_tRht · 2021-11-20
> > **Need to show the results of multiple trials with different random seeds**
> >
> > Showing the results of a single trial is not very appropriate. The results in Table 2 and 3 are important to support the claim regarding the diversity of solutions. Please show the results of multiple trials with different random seeds in Tables 2, 3, 6, 7, and 8.

---

> > > ### Author Response · Authors · 2021-11-21
> > > **thanks for your feedback**
> > >
> > > We would like to emphasize that for simpler scenarios with known local optima, i.e., 4-Goals and Stag-Hunt scenarios, all the results are averaged across multiple seeds (Table 1, Fig 2, and Fig 7). As we have mentioned at the beginning of the experiment section, we *quantitatively* examine the performance of RSPO on these simpler games while using MuJoCo and SMAC as a more challenging testbed for *qualitative* visualization.
> > >
> > > We have conducted multiple experiment repetitions on MuJoCo scenarios with results updated in the paper.
> > >
> > > For the StarCraft domain, we have repeatedly executed RSPO for multiple trials and RSPO could consistently and stably produce diverse behaviors. While for the baselines, due to limited computation resources, we promise to update Table 3 in the camera-ready version. Finally, we want to remark that even from the current numbers, noting that RSPO consistently produces the perfect diversity, all baseline methods perform clearly worse.

---

### Official Review · Reviewer_YXUG · 2021-11-02

**Correctness:** 4
**Technical Novelty And Significance:** 3
**Empirical Novelty And Significance:** 3
**Recommendation:** 8
**Confidence:** 4

**Main Review:**

Overall the paper is of high quality. It is well structured and describes concisely and clearly the work done. The experimental evaluation is appropriate and demonstrates the benefits of the proposed method in a large variety of domains.

While the novelty of this paper is clear from the domain of deep reinforcement learning, the paper does not consider recent approaches that combine QD algorithms and policy gradient algorithms, such as QD-RL[1] or PGA-MAP-Elites[2]. These approaches can generate collections of hundreds of NN policies and would appear as suitable baselines for this paper. One might say that these approaches require a well-defined behaviour descriptor (like MAP-Elites approaches). Yet, it is possible to use distance-based archives (like the unstructured archive in [3]) which only require the definition of a distance function and a threshold value, like in eq.(1).

An aspect of the algorithm that remains unclear to me after looking at the experimental results is if every iteration necessarily leads to a policy added to the collection. Figures 1.a, and 7.b show that it is possible to have fewer strategies than the number of iterations. Is it because the authors listed the possible strategies and they counted how many of these were discovered by the algorithms, or is it that in certain cases, the algorithms would fail to produce a policy that satisfies the hard constraint and thus fail?

Minor comments:
- MAP-Elites belongs to the family of QD algorithms, and there is a typo in its name in the paper.
- "Justin K. Pugh, L. B. Soros, and K. Stanley. Quality diversity: A new frontier for evolutionary computation." appears twice in the reference list.

[1] Cideron, G., Pierrot, T., Perrin, N., Beguir, K., & Sigaud, O. (2020). QD-RL: Efficient Mixing of Quality and Diversity in Reinforcement Learning. arXiv preprint arXiv:2006.08505.
[2] Nilsson, O., & Cully, A. (2021, June). Policy gradient assisted MAP-Elites. In Proceedings of the Genetic and Evolutionary Computation Conference (pp. 866-875).
[3] Cully, A., & Demiris, Y. (2017). Quality and diversity optimization: A unifying modular framework. IEEE Transactions on Evolutionary Computation, 22(2), 245-259.

**Summary Of The Paper:**

In the paper "Continuously Discovering Novel Strategies via Reward-Switching Policy Optimization", the authors introduce a new reinforcement learning algorithm to iteratively form a collection of high-performing and diverse policies. The main idea is that at each iteration of the algorithm, a new policy is optimised so that it maximises the extrinsic reward and satisfy the condition of being different enough from the policy already in the collection. In order to solve this constrained reinforcement learning problem, the authors propose a new loss that effectively rejects the strategies that are similar to already discovered ones and provide an intrinsic reward for pushing these strategies as far as possible before considering the extrinsic reward. They evaluate the benefits of the proposed algorithm on a large set of experiments, including continuous control tasks, and complex multi-agent domains.

**Summary Of The Review:**

Overall the paper is of high quality. It is well structured and describes concisely and clearly the work done. The experimental evaluation is appropriate and demonstrates the benefits of the proposed method in a large variety of domains. I believe the paper could be improved by also considering recent work from the QD literature which could be applied in similar domains while scaling to larger collections.

---

> ### Author Response · Authors · 2021-11-16
> **Author Response to Reviewer YXUG**
>
> Thank you for your appreciation of our work. Please see our detailed explanations below. Our paper has been updated accordingly.
>
> > the paper does not consider recent approaches that combine QD algorithms and policy gradient algorithms
>
> + We appreciate this suggestion.  As an additional baseline, we run PGA-MAP-Elites on the 4-goals and the Escalation environment, where the behavioral descriptors (BDs) can be precisely defined. In particular, we define a 4-dimensional descriptor for the 4-goals environment, i.e., a one-hot vector indicating the ID of the nearest landmark. For the Escalation environment, we use a 1-dimensional descriptor for the Escalation environment, which is a 0-1-normalized value of the cooperation steps within the episode. We set the number of behavior cells (niches) equal to the iteration number in RSPO.
>
> + The average number of different strategies discovered by PGA-MAP-Elites and RSPO across 3 random seeds is shown below.
>
> |Environment|4-goals easy| 4-goals medium| 4-goals hard | Escalation|score in 4-goals hard|
> | ---|---|---|---|---|---|
> |PGA-MAP-Elites|4|1.6|1.2|10|0.98|
> |RSPO|4|4|4|7.7|1.30|
>
>
> Note that although PGA-MAP-Elites outperformed RSPO in the Escalation environment thanks to the informative PD and the large population size (i.e., the archive), RSPO consistently performs better in the 4-goals environments.
>
> + We would also like to discuss some characteristics of evolutionary algorithms and RSPO below:
>
> 1.  Evolutionary algorithms may produce policies that have not yet converged while our objective aims to find diverse *local optima*. We remark that when measuring population diversity, these unconverged policies would contribute a lot even though many of them may have unsatisfying behaviors/returns.
>
> 2. Evolutionary algorithms introduce a substantially larger population size compared with RSPO. The aim of our paper is to *efficiently* discover as many local optima as possible, which induces a small, diverse, and high-performing population. By contrast, evolutionary algorithms evolve over a *large* archive to find several best-performing solutions, which yields many redundant policies.
>
> 3. Defining BDs may not always be feasible in many complex scenarios. For example, in Monster-Hunt, generating BDs requires strong domain knowledge with temporal behavior features. Moreover, in environments like SMAC, we do not even know what kind of behaviors will emerge after training.
>
> 4. Maintaining a large unstructured archive can be particularly computationally expensive for complex environments like SMAC, which require a long training process. It can be challenging to visually evaluate learned behaviors given such a large population.
>
> Finally, we do agree that when an effective and informative BD is available, PGA-MAP-Elites can be a strong candidate, although it may not fit all the scenarios. It could be also beneficial to investigate how to incorporate informative domain prior into the RSPO framework, which we leave as our future work.
>
> + We have added all the results of PGA-MAP-Elites and the discussions into Appendix B.4. Citation is also added to the related work section.
>
>
> > An aspect of the algorithm that remains unclear to me after looking at the experimental results is if every iteration necessarily leads to a policy added to the collection. Figures 1.a, and 7.b show that it is possible to have fewer strategies than the number of iterations.
>
> + We add all the learned policies from every training iteration to the collection for simplicity and count the discovered modes at the very end. Note that as more strategy modes are discovered, RSPO is solving an increasingly challenging constrained optimization problem, so it is indeed possible that an iteration "fails", e.g., some constraints may be violated or the policies may simply converge to previous modes. We empirically observe that these "failure" iterations typically lead to visually indistinguishable behaviors, which would not affect our final evaluation metric. We have clarified this in the revised paper.

---

> > ### Comment · Reviewer_YXUG · 2021-11-19
> > **Thank you for addressing my comments.**
> >
> > I am very happy to see the addition of PGA-MAP-Elites in the baselines. However, I find it strange that these results are not reported alongside the main results of the paper. In particular, for the escalation experiment, PGA-MAP-Elites seems to find more strategies than any other algorithm and yet this is not discussed in the main paper. Actually, the B.4 section is not even mentioned in the main text. I think these results should be properly highlighted in the main text.
> >
> > A few comments on the discussion about Evolutionary algorithms:
> > 1. _Evolutionary algorithms may produce policies that have not yet converged while our objective aims to find diverse local optima. We remark that when measuring population diversity, these unconverged policies would contribute a lot even though many of them may have unsatisfying behaviors/returns._
> >
> > Yes, this is correct, and it would be interesting to study (e.g., in future work) if this is effectively the case or not.
> >
> > 2. _Evolutionary algorithms introduce a substantially larger population size compared with RSPO. The aim of our paper is to efficiently discover as many local optima as possible, which induces a small, diverse, and high-performing population. By contrast, evolutionary algorithms evolve over a large archive to find several best-performing solutions, which yields many redundant policies._
> >
> > While I would agree with this comment in general, this does not seem to be the case in the presented experiments. If I understand correctly, the size of the archive is respectively 4 and 10 cells, which is either smaller or equal to the number of iterations done by RSPO. These numbers are not in my opinion "substantially larger" than what RSPO uses, but very similar and comparable. The difference is that PGA-MAP-Elites trains all the policies simultaneously instead of iteratively like RSPO.
> > Moreover, the claim that this "yields many redundant policies" is not supported by the experimental results.
> >
> > It is also important to note Quality-Diversity algorithms like MAP-Elites and PGA-MAP-Elites are designed to create collections with hundreds or thousands of cells. Having a large number of cells enables these algorithms to empirically compute a "diversity gradient" and thus to explore better. Running PGA-MAP-Elites with 4 and 10 cells is likely to penalise its performance, but this can be justified to make the comparison with RSPO as it considers the same number of policies.
> >
> >
> >
> > 3. _Defining BDs may not always be feasible in many complex scenarios. For example, in Monster-Hunt, generating BDs requires strong domain knowledge with temporal behavior features. Moreover, in environments like SMAC, we do not even known what kind of behaviors will emerge after training._
> >
> > This is true, but several papers already offer solutions to learn in an unsupervised manger the BD [1,2] or directly from in-game images [3]. Yet, I do not believe these have been already applied to environments like SMAC.
> >
> > 4. _Maintaining a large unstructured archive can be particularly computationally expensive for complex environments like SMAC, which require a long training process. It can be challenging to visually evaluate learned behaviors given such a large population._
> >
> > I agree, but again this is not the case in the presented experiment. Conversely, your experiments show that these algorithms can also be applied with a small population and still provide effective solutions.
> >
> > I am happy with the additional explanation on the failed iterations.
> >
> > Here is a small typo I noticed:
> > "itertion" -> "iteration"
> >
> >
> > [1] Cully, A. (2019, July). Autonomous skill discovery with quality-diversity and unsupervised descriptors. In Proceedings of the Genetic and Evolutionary Computation Conference (pp. 81-89).
> >
> > [2] Paolo, G., Laflaquiere, A., Coninx, A., & Doncieux, S. (2020, May). Unsupervised learning and exploration of reachable outcome space. In 2020 IEEE International Conference on Robotics and Automation (ICRA) (pp. 2379-2385). IEEE.
> >
> > [3] Ecoffet, A., Huizinga, J., Lehman, J., Stanley, K. O., & Clune, J. (2021). First return, then explore. Nature, 590(7847), 580-586.

---

> > > ### Author Response · Authors · 2021-11-21
> > > **Author Response 2 to Reviewer YXUG**
> > >
> > > Thanks for your comments. Although somewhat parallel to our focus, it has been a great conversation on the topic of evolutionary methods, which we do learn a lot from but attract less attention in the current RL literature unfortunately. We really appreciate and agree with many of your opinions.
> > >
> > > Regarding the paper, we have largely revised Appendix B.4 and highlighted the study on PGA-MAP-Elites at the beginning of the experiment section. We have incorporated many suggestions into appendix B.4 and made it look more like a self-contained study rather than an individual additional experiment for readers of further interest.
> > >
> > > We would also like to clarify that the success of PGA-MAP-Elites on Escalation largely relies on our best effort of BD design, where each niche precisely corresponds to a range of NEs. By contrast, RSPO directly works on the particularly deceptive environment reward (see Appendix C) without knowing any domain knowledge. As we have discussed in Appendix B.4, we fully agree that evolutionary algorithms can be strong candidates when BD is available. However, it can be problematic to directly compare RSPO and PGA-MAP-Elites without clearly specifying the underlying assumption, so for the best presentation consistency, we decide to primarily present generally-applicable RL methods in the main paper while leaving this study in the appendix.
> > >
> > > A minor comment: we ran PGA-MAP-Elites with 7 cells in the 4-goals environment, the same as baseline methods. We have clarified this in Appendix B.4.
> > >
> > > Finally, we would be also very happy to continue our conversation on some of the mentioned advances in evolutionary methods. From my personal perspective, a fundamental difference between evolutionary methods and RL-based works is that evolutionary methods require a human-designed low-dimensional function (e.g., BD, BC, QD, etc), which typically incorporates domain knowledge to some extent; while a standard RL work generally does not require any domain knowledge. This assumption difference often makes it difficult to directly compare evolutionary methods with RL-based methods in an apple-to-apple manner. Although it is exciting to see recent attempts to use VAEs to learn state representations or use state hashing as descriptors, designing BD for modeling more complex strategy or temporal behaviors is still far from satisfactory (e.g., SMAC). We are definitely happy to see further progress in this direction and do believe many concepts and techniques from evolutionary methods could largely benefit RL works. However, covering all these extension possibilities would be much beyond the scope of a single conference paper.

---

> > > > ### Comment · Reviewer_YXUG · 2021-11-29
> > > > **Response**
> > > >
> > > > Thank you for the additional changes and clarifications made on the paper.
> > > >
> > > > I will update my score to 8.

---

### Official Review · Reviewer_LnVU · 2021-11-03

**Correctness:** 4
**Technical Novelty And Significance:** 2
**Empirical Novelty And Significance:** 3
**Recommendation:** 8
**Confidence:** 3

**Main Review:**

The problem is well-formulated and experiments are validating the claims authors made in the paper. The overall idea of the proposed method makes sense to me.

I have the following concerns and questions regarding this paper. I can adjust my score accordingly after the authors reply to these questions:

(1) My first concern is regarding $\delta$. I appreciate the author’s effort to provide an empirical way of adjusting it, but I didn’t find it intuitive. Such an important parameter requires a better Ablation study. The results of Table 1 shows the occurrence of different strategies by stopping the reward switching. But does it say much? Or could it be just randomly happening? I suggest showing a percentage of each strategy that is discovered, or any better way of investigating this parameter.

(2) My second concern is regarding using the reward prediction error as your reward-driven intrinsic reward. To my understanding, this value may never be accurate, specially at the start of learning $f(.,.,\psi)$. Also does it somehow relate to this paper? https://proceedings.neurips.cc/paper/2018/file/51de85ddd068f0bc787691d356176df9-Paper.pdf

(3) The ordering of the Figures in the experiments could be much better. I believe authors can fix this issue as it affects the readability of paper. E.g. Figure 4 in the main text has been referred to only in the Appendix. Page 8 contains 3 figures, start from 7, then 6, and then 5. The location of Table 1 doesn’t match with the corresponding text. Figure 7 (a) does not clearly show other approaches (PG,DIPG,RND) due to the overlay. Maybe authors can have a small zoom-in box in there?


**Summary Of The Paper:**

This paper presents a Reward-Switching Policy Optimization (RSPO) that discovers diverse policies that are locally optimal and sufficiently different from the existing ones. This is done by adaptive switching among extrinsic and intrinsic rewards that are used for policy learning. RSPO makes use of both the trajectory filtering objective and the exploration objective.

**Summary Of The Review:**

Overall, the paper seems to contribute to the field, but small changes are required before publication.

---

> ### Author Response · Authors · 2021-11-16
> **Author Response to Reviewer LnVU**
>
> We would like to thank you for your appreciation and constructive feedback. We hope our following responses can address your concerns.
>
> > My first concern is regarding $\delta$...I didn’t find it intuitive. Such an important parameter requires a better ablation study.
>
> + Please refer to the sensitivity analysis in Appendix B.3 of the revised version.
>
> > The results of Table 1 show the occurrence of different strategies by stopping the reward switching. But does it say much? Or could it be just randomly happening?
>
> + The results in Table 1 are ablation studies, which suggest that it is necessary to include all the techniques of RSPO in order to discover all the modes in Monster-Hunt. By turning off reward switching, RSPO degenerates to the soft objective of Eq.4 via intrinsic rewards. Although these variants are less powerful than RSPO, they could still discover a few strategy modes because the intrinsic rewards encourage the learning policy to be different from previous references.
>
> > … the reward prediction error..., To my understanding, this value may never be accurate, especially at the start of learning…. Also, does it somehow relate to this paper?
>
> + First, we remark that the reward prediction function w.r.t. $\pi_k$ is separately learned **after** $\pi_k$ converges and **before** the next iteration starts. Hence, whenever the reward prediction is used, it is always well trained.
> + Second, we would like to argue that the accuracy of reward predictions may not be a big issue in practice. It has been a popular approximation technique used in many papers on RL exploration  [1-2]. Empirical evidence suggests that such an approximation scheme can effectively lead to strong exploration performances in various RL applications.
> + Regarding the reference you provided, this paper learns an intrinsic reward for better exploration, while we simply utilize the learned reward-prediction-error as our exploration bonus. It is possible to further improve RSPO with better exploration techniques within each iteration, which is parallel to our current focus. We have included the reference in our revision.
>
> > The ordering of the Figures in the experiments could be much better.
>
> + We apologize for our confusing presentation. We have re-organized the figures and added the missing discussions on Fig 5 in the revised paper (or Fig 4 in our initial draft). Regarding Figure 7 (a), baseline methods indeed make hardly any performance difference.
>
> [1] Liu, X., Jia, H., Wen, Y., Yang, Y., Hu, Y., Chen, Y., ... & Hu, Z. (2021). Unifying Behavioral and Response Diversity for Open-ended Learning in Zero-sum Games. NeuraIPS 2021.
>
> [2] Simmons-Edler, R., Eisner, B., Yang, D., Bisulco, A., Mitchell, E., Seung, S., & Lee, D. (2019). Reward Prediction Error as an Exploration Objective in Deep RL. arXiv preprint arXiv:1906.08189.

---

> > ### Comment · Reviewer_LnVU · 2021-12-01
> > **Thanks for clarifications**
> >
> > I appreciate your response and I checked the updated manuscript.
> > I increase my score to 8.

---

### Author Response · Authors · 2021-11-16
**Revision Comment**

Dear reviewers, we have revised our paper according to your valuable suggestions. We have…
+ Added sensitivity analysis in Appendix B.3
+ Added the additional PGA-MAP-Elites baseline and corresponding discussions in Appendix B.4
+ Clarified some terminologies in the method section
+ Clarified some implementation details in the experiment section and Appendix D.2
+ Added theoretical justifications about the trajectory filtering and the reward switching objective in Appendix G
+ Added more algorithmic discussions in Appendix H
+ Re-organized figures and tables for better readability
+ Fixed some errors and add some citations in the related work section
+ Revised Section 3 with more clarifications
+ Added standard deviation across multiple trials in Table 2 and Table 6

We sincerely hope the revision could address your concerns.

---
Thanks to the suggestions from AC, we have made the theoretical justification (Appendix G) of our method more rigorous. The updated draft can be found on [our project website](https://sites.google.com/view/rspo-iclr-2022) or [here](https://drive.google.com/file/d/1MceCxT37u979ZoB7hWWba_5H79XkA48s/view?usp=sharing).

---

### Decision · Program_Chairs · 2022-01-20

**Decision:**

Accept (Poster)

**Comment:**

In this paper, a new method is proposed to discover diverse policies solving a given task. The key ideas are to (1) learn one policy at a time, with each new policy trying to be different enough from the previous ones, and (2) switch between two rewards on a per-trajectory basis: the "normal" reward on trajectories that are unlikely enough under previoiusly discovered policies, and a "diversity-inducing" reward on trajectories that are too likely (so as to push the policy being learned away from the previous ones). The main benefit of this switching mechanism is to ensure that the new policy will be optimal, because the reward signal isn't "diluted" by the diversity-inducing signal as long as the policy stays far away from the previous ones.

After the discussion period, most reviewers clearly recommended acceptance of the paper. One reviewer remained on the "reject" side though, especially due to an unconvincing theoretical analysis of the method, in spite of several back and forth with authors. I also had my own concerns regarding that part after reading the paper, and further discussions with authors eventualy led to a significant rewrite of the corresponding theorems and proofs. I believe the final version (shared in comments by authors after the dealine for paper revisions) to at least be technically correct, though the relevance of the theory w.r.t. practical usage of the method is still not entirely convincing (e.g., assumptions regarding the number of distinct global optima, and the need for positive rewards).

That being said, in spite of these concerns regarding the practical significance of the theoretical analysis, I believe the paper has a strong enough empirical validation, and the method is (1) simple, (2) intuitively reasonable, (3) original due to the trajectory-switching mechanism, which makes me recommend acceptance.